# When Can We Track Significant Preference Shifts in Dueling Bandits?

**Joe Suk**
Columbia University
joe.suk@columbia.edu

**Arpit Agarwal**
Columbia University
aa4931@columbia.edu

## Abstract

The $K$-armed dueling bandits problem, where the feedback is in the form of noisy pairwise preferences, has been widely studied due its applications in information retrieval, recommendation systems, etc. Motivated by concerns that user preferences/tastes can evolve over time, we consider the problem of *dueling bandits with distribution shifts*. Specifically, we study the recent notion of *significant shifts* [Suk and Kpotufe, 2022], and ask whether one can design an *adaptive* algorithm for the dueling problem with $O(\sqrt{K\tilde{L}T})$ dynamic regret, where $\tilde{L}$ is the (unknown) number of significant shifts in preferences. We show that the answer to this question depends on the properties of underlying preference distributions. Firstly, we give an impossibility result that rules out any algorithm with $O(\sqrt{K\tilde{L}T})$ dynamic regret under the well-studied Condorcet and SST classes of preference distributions. Secondly, we show that SST∩STI is the largest amongst popular classes of preference distributions where it is possible to design such an algorithm. Overall, our results provides an almost complete resolution of the above question for the hierarchy of distribution classes.

## 1 Introduction

The $K$-armed dueling bandits problem has been well-studied in the multi-armed bandits literature [Yue and Joachims, 2011, Yue et al., 2012b, Urvoy et al., 2013, Ailon et al., 2014, Zoghi et al., 2014, 2015a,b, Dudik et al., 2015, Jamieson et al., 2015, Komiyama et al., 2015, 2016, Ramamohan et al., 2016, Chen and Frazier, 2017, Saha and Gaillard, 2022, Agarwal et al., 2022]. In this problem, on each trial $t \in [T]$, the learner pulls a *pair* of arms and observes *relative feedback* between these arms indicating which arm was preferred. The feedback is typically stochastic, drawn according to a pairwise preference matrix $\mathbf{P} \in [0, 1]^{K \times K}$, and the regret measures the 'sub-optimality' of arms with respect to a 'best' arm.

This problem has many applications, e.g. information retrieval, recommendation systems, etc, where relative feedback between arms is easy to elicit, while real-valued feedback is difficult to obtain or interpret. For example, a central task for information retrieval algorithms is to output a ranked list of documents in response to a query. The framework of online learning has been very useful for automatic parameter tuning, i.e. finding the best parameter(s), for such retrieval algorithms based on user feedback [Liu, 2009]. However, it is often difficult to get numerical feedback for an individual list of documents. Instead, one can (implicitly) compare two lists of documents by interleaving them and observing the relative number of clicks [Radlinski et al., 2008]. The availability of these pairwise comparisons allows one to tune the parameters of retrieval algorithms in real-time using the framework of dueling bandits.

However, in many such applications that rely on user generated preference feedback, there are practical concerns that the tastes/beliefs of users can change over time, resulting in a dynamically

37th Conference on Neural Information Processing Systems (NeurIPS 2023).

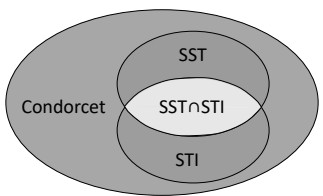

Figure 1: The hierarchy of distribution classes. The dark region is where $O(\sqrt{K\tilde{L}T})$ dynamic regret is not achievable, whereas the light region indicates achievablility (e.g., by our Algorithm 1).

changing preference distribution. Motivated by these concerns, we consider the problem of *switching dueling bandits* (or non-stationary dueling bandits), where the pairwise preference matrix $\mathbf{P}_t$ changes an unknown number of times over $T$ rounds. The performance of the learner is evaluated using *dynamic regret* where sub-optimality of arms is calculated with respect to the current 'best' arm.

Saha and Gupta [2022] first studied this problem and provided an algorithm that achieves a nearly optimal (up to $\log$ terms) *dynamic regret* of $\tilde{O}(\sqrt{KLT})$ where $L$ is the total number of *shifts* in the preference matrix, i.e., the number of times $\mathbf{P}_t$ differs from $\mathbf{P}_{t+1}$. However, this result requires algorithm knowledge of $L$. Alternatively, the algorithm of Saha and Gupta [2022] can be tuned to achieve a dynamic regret rate (also nearly optimal) $\tilde{O}(V_T^{1/3}K^{1/3}T^{2/3})$ in terms of the total-variation of change in preferences $V_T$ over $T$ total rounds. This is similarly limited by requiring knowledge of $V_T$.

On the other hand, recent works on the switching MAB problem show it is not only possible to design *adaptive* algorithms with $\tilde{O}(\sqrt{KLT})$ dynamic regret without knowledge of the underlying environment [Auer et al., 2019], but also possible to achieve a much better bound of $\tilde{O}(\sqrt{K\tilde{L}T})$ where $\tilde{L} \ll L$ is the number of *significant shifts* [Suk and Kpotufe, 2022]. Specifically, a shift is significant when there is no 'safe' arm left to play, i.e., every arm has, on some interval $[s_1, s_2]$, regret order $\Omega(\sqrt{s_2 - s_1})$. Such a weaker measure of non-stationarity is appealing as it captures the changes in best-arm which are most severe, and allows for more optimistic regret rates over the previously known $\sqrt{KLT} \wedge (V_T K)^{1/3}T^{2/3}$.

Very recently, Buening and Saha [2022] considered an analogous notion of significant shifts for switching dueling bandits under the SST∩STI[1] assumption. They gave an algorithm that achieves a dynamic regret of $\tilde{O}(K\sqrt{\tilde{L}T})$, where $\tilde{L}$ is the (unknown) number of *significant shifts*. However, their algorithm estimates $\Omega(K^2)$ pairwise preferences, and hence, suffers from a sub-optimal dependence on $K$.

In this paper we consider the goal of designing *optimal* algorithms for switching dueling bandits whose regret depends on the number of *significant* shifts $\tilde{L}$. We ask the following question:

**Question.** *Is it possible to achieve a dynamic regret of $O(\sqrt{K\tilde{L}T})$ without knowledge of $\tilde{L}$?*

We show that the answer to this question depends on conditions on the preference matrices. Specifically, we consider several well-studied conditions from the dueling bandits literature, and give an almost complete resolution of the achievability of $O(\sqrt{K\tilde{L}T})$ dynamic regret under these conditions.

### 1.1 Our Contributions

We first consider the classical *Condorcet winner* (CW) condition where, at each time $t \in [T]$, there is a 'best' arm under the preference $\mathbf{P}_t$ that stochastically beats every other arm. Such a winner arm is a benchmark in defining the aforementioned dueling *dynamic regret*. Our first result shows that, even under the CW condition, it is in general impossible to achieve $O(\sqrt{K\tilde{L}T})$ dynamic regret.

**Theorem 1.** *(Informal) There is a family of instances $\mathcal{F}$ under Condorcet where all shifts are non-significant, i.e. $\tilde{L} = 0$, but no algorithm can achieve $o(T)$ dynamic regret uniformly over $\mathcal{F}$.*

Note that in the case when $\tilde{L} = 0$, one would ideally like to achieve a dynamic regret of $O(\sqrt{KT})$. The above theorem shows that, under the Condorcet condition when $\tilde{L} = 0$, not only is it impossible

---

[1]SST∩STI imposes a linear ordering over arms and two well-known conditions on the preference matrices: strong stochastic transitivity (SST) and stochastic triangle inequality (STI).

to achieve $O(\sqrt{KT})$ regret, it is even impossible to achieve $O(T^\alpha)$ regret for any $\alpha < 1$. Hence, this rules out the possibility of an algorithm whose regret under this condition is sublinear in $\tilde{L}$ and $T$.

The proof of the above theorem relies on a careful construction where, at each time $t$, the preference $\mathbf{P}_t$ is chosen uniformly at random from two different matrices $\mathbf{P}^+$ and $\mathbf{P}^-$. These matrices have different 'best' arms but there is a unique *safe arm* in both. However, it is impossible to identify this safe arm as all observed pairwise preferences are $\mathrm{Ber}(\frac{1}{2})$ over the randomness of the environment. Moreover, the theorem gives two different constructions (one ruling out SST and one STI) which together rule out all preference classes outside of SST∩STI. Our second result shows that the desired regret $\sqrt{K\tilde{L}T}$ is in fact achievable (adaptively) under SST∩STI.

**Theorem 2.** *(Informal) There is an algorithm that achieves a dynamic regret of* $\tilde{O}(\sqrt{K\tilde{L}T})$ *under* SST∩STI *without requiring knowledge of* $\tilde{L}$.

Figure 1 gives a summary of our results. Note that in stationary dueling bandits there is no separation in the regret achievable under the CW vs. SST∩STI conditions, i.e. $O(\sqrt{KT})$ is the minimax optimal regret rate under both conditions [Saha and Gaillard, 2022]. However, our results show that in the non-stationary setting with regret in terms of significant shifts, there is a separation in adaptively achievable regret.

**Key Challenge and Novelty in Regret Upper Bound:** To contrast, the recent work of Buening and Saha [2022] only attains $\tilde{O}(K\sqrt{\tilde{L}T})$ dynamic regret under SST∩STI due to inefficient exploration of arm pairs. Our more challenging goal of obtaining the optimal dependence on $K$ introduces key difficulties in algorithmic design. In fact, even in the classical stochastic dueling bandit problem with SST∩STI, most existing results that achieve $O(\sqrt{KT})$ regret require identifying a coarse ranking over arms to avoid suboptimal exploration of low ranked arms [Yue et al., 2012a, Yue and Joachims, 2011]. However, in the non-stationary setting, ranking the arms meaningfully is difficult as the true ordering of arms may change (insignificantly) at all rounds. Our main algorithmic innovation is to bypass the task of ranking arms and instead directly focus on minimizing the cumulative regret of played arms. This entails a new rule for selecting "candidate" arms based on cumulative regret that may be of independent interest.

### 1.2 Related Work

**Dueling bandits.** The stochastic dueling bandits problem and its variants have been studied widely (see Sui et al. [2018] for a comprehensive survey). This problem was first proposed by Yue et al. [2012b], who provide an algorithm achieving instance-dependent $O(K \log T)$ regret under the SST∩STI condition. Yue and Joachims [2011] also studied this problem under the SST∩STI condition and gave an algorithm that achieves optimal instance-dependent regret. Urvoy et al. [2013] studied this problem under the Condorcet winner condition and achieved an instance-dependent $O(K^2 \log T)$ regret bound, which was further improved by Zoghi et al. [2014] and Komiyama et al. [2015] to $O(K^2 + K \log T)$. Finally, Saha and Gaillard [2022] showed that it is possible to achieve an optimal instance-dependent bound of $O(K \log T)$ and instance-independent bound of $O(\sqrt{KT})$ under the Condorcet condition. More general notions of winners such as Borda winner [Jamieson et al., 2015], Copeland winner [Zoghi et al., 2015a, Komiyama et al., 2016, Wu and Liu, 2016], and von Nuemann winner [Dudik et al., 2015] have also been considered. However, these works only consider the stationary setting whereas we consider the non-stationary setting.

There has also been work on adversarial dueling bandits [Saha et al., 2021, Gajane et al., 2015], however, these works only consider static regret against the 'best' arm in hindsight and whereas we consider the harder dynamic regret. Other than the two previously mentioned works [Gupta and Saha, 2022, Buening and Saha, 2022], the only other work on switching dueling bandits is Kolpaczki et al. [2022], whose procedures require knowledge of non-stationarity and only consider the weaker measure of non-stationarity $L$ counting all changes in the preferences.

**Non-stationary multi-armed bandits.** Multi-armed bandits with changing rewards was first considered in the adversarial setup by Auer et al. [2002], where a version of EXP3 was shown to attain optimal dynamic regret $\sqrt{KLT}$ when properly tuned using the number $L$ of changes in the rewards. Later works established similar (non-adaptive) guarantees in this so-called *switching bandit* problem via procedures inspired by stochastic bandit algorithms [Garivier and Moulines, 2011, Kocsis

and Szepesvári, 2006]. More recent works [Auer et al., 2018, 2019, Chen et al., 2019] established the first adaptive and optimal dynamic regret guarantees, without requiring knowledge of the number of changes. An alternative parametrization of switching bandits, via a total-variation quantity, was introduced in Besbes et al. [2014] with minimax rates quantified therein and adaptive rates attained in Chen et al. [2019]. Yet another characterization, in terms of the number of best arm switches $S$ was studied in Abbasi-Yadkori et al. [2022], establishing an adaptive regret rate of $\sqrt{SKT}$. Around the same time, Suk and Kpotufe [2022] introduced the aforementioned notion of *significant shifts* and adaptively achieved rates of the form $\sqrt{K\tilde{L}T}$ in terms of $\tilde{L}$ significant shifts in rewards.

## 2   Problem Formulation

We consider non-stationary dueling bandits with $K$ arms and time-horizon $T$. At round $t \in [T]$, the pairwise preference matrix is denoted by $\mathbf{P}_t \in [0,1]^{K \times K}$, where the $(i,j)$-th entry $P_t(i,j)$ encodes the likelihood of observing a preference for arm $i$ in a direct comparison with arm $j$. The preference matrix may change arbitrarily from round to round. At round $t$, the learner selects a pair of actions $(i_t, j_t) \in [K] \times [K]$ and observes the feedback $O_t(i_t, j_t) \sim \mathrm{Ber}(P_t(i_t, j_t))$ where $P_t(i_t, j_t)$ is the underlying preference of arm $i_t$ over $j_t$. We define the pairwise gaps $\delta_t(i,j) := P_t(i,j) - 1/2$.

**Conditions on Preference Matrix.** We consider two different conditions on preference matrices: (1) the Condorcet winner (CW) condition and (2) the strong stochastic transitivity (SST) and stochastic triangle inequality (STI), formalized below.

**Definition 1.** *(CW condition) At each round $t$, there is a **Condorcet winner** arm, denoted by $a_t^*$, such that $\delta_t(a_t^*, a) \geq 0$ for all $a \in [K] \backslash \{a_t^*\}$. Note that $a_t^*$ need not be unique.*

**Definition 2.** *(SST∩STI condition) At each round $t$, there exists a total ordering on arms, denoted by $\succ_t$, and $\forall i \succeq_t j \succeq_t k$:*

*(a) $\delta_t(i,k) \geq \max\{\delta_t(i,j), \delta_t(j,k)\}$ (SST).*

*(b) $\delta_t(i,k) \leq \delta_t(i,j) + \delta_t(i,k)$ (STI).*

It's easy to see that the SST condition implies the CW condition as $\delta_t(i,j) \geq \delta_t(i,i) = 0$ for any $i \succ_t j$. Hence, the highest ranked item under $\succ_t$ in Definition 2 is the CW $a_t^*$. We emphasize here that the CW in Definition 1 and the total ordering on arms in Definition 2 can change at each round, even while such unknown changes in preference may not be counted as significant (see below).

**Regret Notion.** Our benchmark is the *dynamic regret* to the sequence of Condorcet winner arms:

$$\mathrm{DR}(T) := \sum_{t=1}^{T} \frac{\delta_t(a_t^*, i_t) + \delta_t(a_t^*, j_t)}{2}.$$

Here, the regret of an arm $i$ is defined in terms of the preference gap $\delta_t(a_t^*, i)$ between the winner arm $a_t^*$ and $i$, and the regret of the pair $(i_t, j_t)$ is the average regret of individual arms $i_t$ and $j_t$. Note the this regret is well-defined under both Condorcet and SST∩STI conditions due to the existence of a unique 'best' arm $a_t^*$, and is non-negative due to the fact that $\delta_t(a_t^*, i) \geq 0$ for all $i \in [K]$.

**Measure of Non-Stationarity.** We first recall the notion of Significant Condorcet Winner Switches from Buening and Saha [2022], which captures only the switches in $a_t^*$ which are severe for regret. Throughout the paper, we'll also refer to these as *significant shifts* for brevity.

**Definition 3** (Significant CW Switches). *Define an arm $a$ as having **significant regret** over $[s_1, s_2]$ if*

$$\sum_{s=s_1}^{s_2} \delta_s(a_s^*, a) \geq \sqrt{K \cdot (s_2 - s_1)}. \tag{1}$$

*We then define **significant CW switches** recursively as follows: let $\tau_0 = 1$ and define the $(i+1)$-th significant CW switch $\tau_{i+1}$ as the smallest $t > \tau_i$ such that for each arm $a \in [K]$, $\exists [s_1, s_2] \subseteq [\tau_i, t]$ such that arm $a$ has significant regret over $[s_1, s_2]$. We refer to the interval of rounds $[\tau_i, \tau_{i+1})$ as a **significant phase**. Let $\tilde{L}$ be the number of significant CW switches elapsed in $T$ rounds.*

**Notation.** *To ease notation, we'll conflate the closed, open, and half-closed intervals of real numbers $[a,b]$, $(a,b)$, and $[a,b)$, with the corresponding rounds contained therein, i.e. $[a,b] \equiv [a,b] \cap \mathbb{N}$.*

# 3 Hardness of Significant Shifts in the Condorcet Winner Setting

We first consider regret minimization in an environment with no significant shift in $T$ rounds. Such an environment admits a *safe arm* $a^\sharp$ which does not incur significant regret throughout play. Our first result shows that, under the Condorcet condition, it is not possible to distinguish the identity of $a^\sharp$ from other unsafe arms, which will in turn make sublinear regret impossible.

**Theorem 3.** *For each horizon $T$, there exists a finite family $\mathcal{F}$ of switching dueling bandit environments with $K = 3$ that satisfies the Condorcet winner condition (Definition 1) with $\tilde{L} = 0$ significant shifts. The worst-case regret of any algorithm on an environment $\mathcal{E}$ in this family is lower bounded as*

$$\sup_{\mathcal{E} \in \mathcal{F}} \mathbb{E}_{\mathcal{E}}\left[\mathrm{DR}(T)\right] \geq T/8.$$

*Proof.* (sketch; details found in Appendix B) Letting $\epsilon \ll 1/\sqrt{T}$, consider the preference matrices:

$$\mathbf{P}^+ := \begin{pmatrix} 1/2 & 1/2 + \epsilon & 1 \\ 1/2 - \epsilon & 1/2 & 1/2 + \epsilon \\ 0 & 1/2 - \epsilon & 1/2 \end{pmatrix}, \mathbf{P}^- := \begin{pmatrix} 1/2 & 1/2 - \epsilon & 0 \\ 1/2 + \epsilon & 1/2 & 1/2 - \epsilon \\ 1 & 1/2 + \epsilon & 1/2 \end{pmatrix}.$$

In $\mathbf{P}^+$, arm 1 is the Condorcet winner and $1 \succ 2 \succ 3$, whereas in $\mathbf{P}^-$, 3 is the winner with $3 \succ 2 \succ 1$. Let an oblivious adversary set $\mathbf{P}_t$ at round $t$ to one of $\mathbf{P}^+$ and $\mathbf{P}^-$, uniformly at random, inducing an environment where arm 2 remains safe for $T$ rounds. Then, any algorithm will, over the randomness of the adversary, observe $O_t(i_t, j_t) \sim \mathrm{Ber}(1/2)$ *no matter the choice of arms* $(i_t, j_t)$ *played*, by the symmetry of $\mathbf{P}^+, \mathbf{P}^-$. Thus, it is impossible to distinguish arms, which implies linear regret by standard Pinsker's inequality arguments. In particular, even a strategy playing arm 2 every round fails as arm 2 is unsafe in another (indistinguishable) setup with arms 1 and 2 switched in $\mathbf{P}^+, \mathbf{P}^-$. □

**SST and STI Both Needed To Learn Significant Shifts.** The preferences $\mathbf{P}^+, \mathbf{P}^-$ in the above proof violate STI but satisfy SST, whereas another construction using preferences $\mathbf{P}^+, \mathbf{P}^-$ which violate SST but satisfy STI also works in the proof (see Remark 2 in Appendix B). This shows that sublinear regret is impossible outside of the class SST∩STI (visualized in Figure 1).

**Remark 1.** *Note the lower bound of Theorem 3 does not violate the established upper bounds $\sqrt{\tilde{L}T}$ and $V_T^{1/3}T^{2/3}$ scaling with $L$ changes in the preference matrix or total variation $V_T$ [Gupta and Saha, 2022]. Our construction in fact uses $L = \Omega(T)$ changes in the preference matrix and $V_T = \Omega(T)$ total variation. Furthermore, the regret upper bound $\sqrt{ST}$, in terms of $S$ changes in Condorcet winner, of Buening and Saha [2022] is not contradicted either, for $S = \Omega(T)$.*

# 4 Dynamic Regret Upper Bounds under SST/STI

Acknowledging that significant shifts are hard outside of the class SST∩STI, we now turn our attention to the achievability of $\sqrt{K\tilde{L}T}$ regret[2] in the SST∩STI setting. Our main result is an optimal dynamic regret upper bound attained *without knowledge of the significant shift times or the number of significant shifts*. Up to log terms, this is the first dynamic regret upper bound with optimal dependence on $T$, $\tilde{L}$, and $K$.

**Theorem 4.** *Suppose SST and STI hold (see Definition 2). Let $\{\tau_i\}_{i=0}^{\tilde{L}}$ denote the unknown significant shifts of Definition 3. Then, for some constant $C_0 > 0$, Algorithm 1 has expected dynamic regret*

$$\mathbb{E}[\mathrm{DR}(T)] \leq C_0 \log^3(T) \sum_{i=0}^{\tilde{L}} \sqrt{K \cdot (\tau_{i+1} - \tau_i)},$$

*and using Jensen's inequality, this implies a regret rate of $C_0 \log^3(T)\sqrt{K \cdot (\tilde{L} + 1) \cdot T}$.*

---

[2]The lower bound construction of Saha and Gupta [2022] in fact uses $\Omega(L)$ significant shifts so that the $\sqrt{L \cdot \tilde{L}}$ rate is in fact minimax optimal

In fact, this regret rate can be transformed to depend on the *Condorcet winner variation* introduced in Buening and Saha [2022] and the *total variation* quantities introduced in Gupta and Saha [2022] and inspired by the total-variation quantity from non-stationary MAB [Besbes et al., 2014]. The following corollary is shown using just the definition of the non-stationarity measures.

**Corollary 5** (Regret in terms of CW Variation). *Let* $V_T := \sum_{t=2}^{T} \max_{a \in [K]} |P_t(a_t^*, a) - P_{t-1}(a_t^*, a)|$ *be the unknown Condorcet winner variation. Using the same notation of Theorem 4: Algorithm 1 has expected dynamic regret*

$$\mathbb{E}[\mathrm{DR}(T)] \leq C_0 \log^3(T) \left( \sqrt{KT} + (KV_T)^{1/3} T^{2/3} \right).$$

## 5 Algorithm

At a high level, the strategy of recent works on non-stationary multi-armed bandits [Chen et al., 2019, Wei and Luo, 2021, Suk and Kpotufe, 2022] is to first design a suitable base algorithm and then use a meta-algorithm to randomly schedule different instances of this base algorithm at variable durations across time. The key idea is that unknown time periods of significant regret can be detected fast enough with the right schedule. In order to accurately identify significant shifts, the base algorithm in question should be robust to all non-significant shifts. In the multi-armed bandit setting, a variant of the classical successive elimination algorithm [Even-Dar et al., 2006] possesses such a guarantee [Allesiardo et al., 2017], and serves as a base algorithm in Suk and Kpotufe [2022].

### 5.1 Difficulty of Efficient Exploration of Arms.

In the non-stationary dueling problem, a natural analogue of successive elimination is to uniformly explore the arm-pair space $[K] \times [K]$ and eliminate arms based on observed comparisons [Urvoy et al., 2013]. The previous work [Theorem 5.1 of Buening and Saha, 2022] employs such a strategy as a base algorithm. However, such a uniform exploration approach incurs a large estimation variance of $K^2$, which enters into the final regret bound of $K\sqrt{T \cdot \tilde{L}}$. Thus, smarter exploration strategies are needed to obtain $\sqrt{K}$ dependence.

In the stationary dueling bandit problem with SST∩STI, such efficient exploration strategies have long been known: namely, the Beat-The-Mean algorithm [Yue and Joachims, 2011] and the Interleaved Filtering (IF) algorithm [Yue et al., 2012a]. We highlight that these existing algorithms aim to learn the ordering of arms, i.e., arms are ruled out roughly in the same order as their true underlying ordering. This fact is crucial to attaining the optimal dependence in $K$ in their regret analyses, as the higher ranked arms must be played more often against other arms to avoid the $K^2$ cost of exploration.

However, in our setting, adversarial but non-significant changes in the ordering of arms could force perpetual exploration of lowest-ranked arms. This suggests that learning an ordering should not be a subtask of our desired dueling base algorithm. Rather, the algorithm should prioritize minimizing its own regret over time. Keeping this intuition in mind, we introduce an algorithm called **SW**itching **I**nterleaved **F**il**T**ering (SWIFT) (see Algorithm 2 in Section 5.2) which directly tracks regret and avoids learning a fixed ordering of arms.

**A new idea for switching candidate arms.** A natural idea that is common to many dueling bandit algorithms (including IF) is to maintain a *candidate arm* $\hat{a}$ which is always played at each round, and serves as a reference point for partially ordering other arms in contention. If the current candidate is beaten by another arm then a new candidate is chosen, and this process quickly converges to the best arm. Since the ordering of arms may change at each round, any such rule that relies on a fixed ordering is deemed to fail in our setting. Our procedure does not rely on such a fixed ordering over arms, but instead tracks the aggregate regret $\sum_t \delta_t(a, \hat{a}_t)$ of the *changing sequence of candidate arms* $\{\hat{a}_t\}_t$ to another fixed arm $a$. Crucially, the candidate arm $\hat{a}_s$ is always played at round $s$ and so the history of candidate arms $\{\hat{a}_s\}_{s \leq t}$ is fixed at a round $t$. This fact allows us to estimate the quantity $\sum_{s=1}^{t} \delta_s(a, \hat{a}_s)$ using importance-weighting at $\sqrt{K \cdot t}$ rates via martingale concentration. An algorithmic *switching criterion* then switches the candidate arm $\hat{a}_t$ to any arm $a$ dominating the sequence $\{\hat{a}_s\}_{s \leq t}$ over time, i.e., $\sum_{s=1}^{t} \delta_s(a, \hat{a}_s) \gg \sqrt{K \cdot t}$. This simple, yet powerful, idea immediately gives us control of the regret of the candidate sequence $\{\hat{a}_t\}_t$ which allows us to bypass the ranking-based arguments of vanilla IF and Beat-The-Mean. It also allows us to simultaneously bound the regret of a sub-optimal arm $a$ against the sequence of candidate arms $\sum_{s=1}^{t} \delta_s(\hat{a}_s, a)$.

## 5.2 Switching Interleaved Filtering (SWIFT)

SWIFT at round $t$ compares a *candidate arm* $\hat{a}_t$ with an arm $a_t$ (chosen uniformly at random) from an *active arm set* $\mathcal{A}_t$. Additionally, SWIFT maintains estimates $\hat{\delta}_t(\hat{a}_t, a)$ of $\delta_t(\hat{a}_t, a)$ which are used to (1) evict active arms $a \in \mathcal{A}_t$ and (2) *switch* the candidate arm $\hat{a}_{t+1}$ for the next round.

**Estimators and Eviction/Switching Criteria.**   Let $\mathcal{A}_t$ be the active arm set at round $t$. Let

$$\hat{\delta}_t(\hat{a}_t, a) := |\mathcal{A}_t| \cdot O_t(\hat{a}_t, a) \cdot \mathbf{1}\{(i_t, j_t) = (\hat{a}_t, a)\} - 1/2, \tag{2}$$

which is an unbiased estimator of the gap $\delta_t(\hat{a}_t, a)$ when $a \in \mathcal{A}_t$. We *evict* an active arm $a$ from $\mathcal{A}_t$ at round $t$ if for some constant $C > 0$[3] and rounds $s_1 < s_2 \le t$:

$$\sum_{s=s_1}^{s_2} \hat{\delta}_s(\hat{a}_s, a) \ge C \log(T) \sqrt{K \cdot (s_2 - s_1) \vee K^2}, \tag{3}$$

where $\hat{\delta}_s(a, \hat{a}_s) := -\hat{\delta}_s(\hat{a}_s, a)$. Next, we switch the next candidate arm $\hat{a}_{t+1} \leftarrow a$ to another arm $a \in \mathcal{A}_t$ at round $t$ if for some round $s_1 < t$:

$$\sum_{s=s_1}^{t} \hat{\delta}_s(a, \hat{a}_s) \ge C \log(T) \sqrt{K \cdot (t - s_1) \vee K^2}. \tag{4}$$

SWIFT is formally shown in Algorithm 2, defined for generic start time $t_{\text{start}}$ and duration $m_0$ so as to allow for recursive calls in our meta-algorithm framework.

## 5.3 Non-Stationary Algorithm (METASWIFT)

---

**Algorithm 1:** **M**eta-**E**limination while **T**racking **A**rms in SWIFT (METASWIFT)

---
**Input:** horizon $T$.
1 **Initialize:** round count $t \leftarrow 1$.
2 **Episode Initialization (setting global variables $t_\ell, \mathcal{A}_{\text{master}}, B_{s,m}$):**
3      $t_\ell \leftarrow t$. ;                           `// `$t_\ell$` indicates start of `$\ell$`-th episode.`
4      $\mathcal{A}_{\text{master}} \leftarrow [K]$ ;                              `// Master active arm set.`
5      For each $m = 2, 4, \dots, 2^{\lceil \log(T) \rceil}$ and $s = t_\ell + 1, \dots, T$:
6          Sample and store $B_{s,m} \sim \text{Bernoulli}\left(\frac{1}{\sqrt{m \cdot (s - t_\ell)}}\right)$. ;      `// Set replay schedule.`

7 Run Base-Alg$(t_\ell, T + 1 - t_\ell)$.
8 **if** $t < T$ **then** restart from Line 2 (i.e. start a new episode). ;

---

For the non-stationary setting with multiple (unknown) significant shifts, we run SWIFT as a base algorithm at randomly scheduled rounds and durations.

Our algorithm, dubbed METASWIFT and found in Algorithm 1, operates in *episodes*, starting each episode by playing a *base algorithm* instance of SWIFT. A running base algorithm *activates* its own base algorithms of varying durations (Line 8 of Algorithm 2), called *replays* according to a random schedule decided by the Bernoulli's $B_{s,m}$ (see Line 6 of Algorithm 1). We refer to the (unique) base algorithm playing at round $t$ as the *active base algorithm*.

**Global Variables.**   The *active arm set* $\mathcal{A}_t$ is pruned by the active base algorithm at round $t$, and globally shared between all running base algorithms. In addition, all other variables, i.e. the $\ell$-th episode start time $t_\ell$, round count $t$, schedule $\{B_{s,m}\}_{s,m}$, and candidate arm $\hat{a}_t$ (and thus the quantities $\delta_t(\hat{a}_t, a)$) are shared between base algorithms. Thus, while active, each Base-Alg can switch the candidate arm (4) and evict arms (3) over all intervals $[s_1, s_2]$ elapsed since it began.

Note that only one base algorithm (the active one) can edit $\mathcal{A}_t$ and set the candidate arm $\hat{a}_t$ at round $t$, while other base algorithms can access these global variables at later rounds. By sharing these global

---
[3] The constant $C > 0$ does not depend on $T$, $K$, or $\tilde{L}$, and a suitable value can be derived from the regret analysis.

---

**Algorithm 2:** Base-Alg($t_{\text{start}}, m_0$): SWIFT starting at $t_0$ and running $m_0$ rounds

---

**Input**: starting round $t_{\text{start}}$, scheduled duration $m_0$.

1 **Initialize (Global) Variables**: $t \leftarrow t_{\text{start}}$, $\mathcal{A}_t \leftarrow [K]$, $\hat{a}_t \leftarrow \text{Unif}\{[K]\}$.
2 **while** $t \leq T$ **do**
3      Select a random arm $a_t \in \mathcal{A}_t$ with probability $1/|\mathcal{A}_t|$ and play $(\hat{a}_t, a_t)$.
4      Let $\mathcal{A}_{\text{current}} \leftarrow \mathcal{A}_t$. ;           `// Save current active arm set` $\mathcal{A}_t$ `(global variable).`
5      Increment $t \leftarrow t + 1$.
6      **if** $\exists m$ such that $B_{t,m} > 0$ **then**       `/* See Algorithm 1 for definition of` $B_{s,m}$ `*/`
7          Let $m := \max\{m \in \{2, 4, \ldots, 2^{\lceil \log(T) \rceil}\} : B_{m,t} > 0\}$. ;     `// Set replay length.`
8          Run Base-Alg($t, m$). ;                  `// Replay interrupts.`
9      **if** $t > t_{\text{start}} + m_0$ **then** RETURN. ;
10     Evict bad arms:
11       $\mathcal{A}_t \leftarrow \mathcal{A}_{\text{current}} \backslash \{a \in [K] : \exists$ rounds $[s_1, s_2] \subseteq [t_{\text{start}}, t)$ s.t. (3) hold$\}$.
12       $\mathcal{A}_{\text{master}} \leftarrow \mathcal{A}_{\text{master}} \backslash \{a \in [K] : \exists$ rounds $[s_1, s_2] \subseteq [t_\ell, t)$ s.t. (3) hold$\}$.
13     **if** (4) holds for some arm $a \in \mathcal{A}_t$ **then**
14       Switch candidate arm: $\hat{a}_t \leftarrow a$. ;       `// Set candidate arm` $\hat{a}_t$ `(global variable).`
15     **else**
16       $\hat{a}_t \leftarrow \hat{a}_{t-1}$.
17     **Restart criterion:** if $\mathcal{A}_{\text{master}} = \emptyset$ **then** RETURN.;
18 RETURN.

---

variables, any replay can trigger a new episode: every time an arm is evicted by a replay, it is also evicted from the *master arm set* $\mathcal{A}_{\text{master}}$, tracking arms' regret throughout the entire episode. A new episode is triggered when $\mathcal{A}_{\text{master}}$ becomes empty, i.e., there is no *safe* arm left to play.

## 6 Regret Analysis

### 6.1 Regret of METASWIFT over Significant Phases

Now, we turn to sketching the proof of Theorem 4. Full details are found in Appendix C.

**Decomposing the Regret.** Let $a_t^\sharp$ denote the *last safe arm* at round $t$, or the last arm to incur significant regret in the unique phase $[\tau_i, \tau_{i+1})$ containing round $t$. Then, we can decompose the dynamic regret around this safe arm using SST and STI (i.e., using Lemma 8 twice) as:

$$\sum_{t=1}^{T} \delta_t(a_t^*, \hat{a}_t) + \delta_t(a_t^*, a_t) \leq 6 \sum_{t=1}^{T} \delta_t(a_t^*, a_t^\sharp) + 3 \sum_{t=1}^{T} \delta_t(a_t^\sharp, \hat{a}_t) + \sum_{t=1}^{T} \delta_t(\hat{a}_t, a_t),$$

where we recall that $a_t \in \mathcal{A}_t$ is the other arm played (Line 3 of Algorithm 2). Next, the first sum on the above RHS is order $\sum_{i=1}^{\tilde{L}} \sqrt{K \cdot (\tau_i - \tau_{i-1})}$ as the last safe arm $a_t^\sharp$ does not incur significant regret on $[\tau_i, \tau_{i+1})$. So, it remains to bound the last two sums on the RHS above.

**Episodes Align with Significant Phases.** We claim that a new episode is triggered only if there a significant shift occurs (Lemma 11). This follows from our eviction criteria (3) with Freedman's inequality for martingale concentration (Lemma 9). Then, acknowledging episodes roughly align with significant phases, we turn our attention to bounding the remaining regret in each episode.

**Bounding Regret of an Episode.** Let $t_\ell$ be the start of the $\ell$-th episode of METASWIFT. Then, our goal is to show for all $\ell \in [\hat{L}]$ (where $\hat{L}$ is the random number of episodes used by the algorithm):

$$\max\left\{ \mathbb{E}\left[ \sum_{t=t_\ell}^{t_{\ell+1}-1} \delta_t(a_t^\sharp, \hat{a}_t) \right], \mathbb{E}\left[ \sum_{t=t_\ell}^{t_{\ell+1}-1} \delta_t(\hat{a}_t, a_t) \right] \right\} \lesssim \sum_{i \in [\tilde{L}+1] : [\tau_{i-1}, \tau_i) \cap [t_\ell, t_{\ell+1}) \neq \emptyset} \sqrt{K \cdot (\tau_i - \tau_{i-1})},$$

$$(5)$$

where the RHS sum above is over the significant phases $[\tau_{i-1}, \tau_i)$ overlapping episode $[t_\ell, t_{\ell+1})$. Summing over episodes $\ell \in [\hat{L}]$ will then yield the desired total regret bound by our earlier observation that the episodes align with significant phases (see Lemma 11).

**Bounding Regret of Active Arms to Candidate Arms.** Bounding $\sum_{t=t_\ell}^{t_{\ell+1}-1} \delta_t(\hat{a}_t, a_t)$ follows in a similar manner as Appendix B.1 of Suk and Kpotufe [2022]. First, observe by concentration (Lemma 9) the eviction criterion (3) bounds the sums $\sum_{t=s_1}^{s_2} \delta_t(\hat{a}_t, a)$ over intervals $[s_1, s_2]$ where $a$ is active. Then, accordingly, we further partition the episode rounds $[t_\ell, t_{\ell+1})$ into different intervals distinguishing the unique regret contributions of different active arms from varying base algorithms, on each of which we can relate the regret to our eviction criterion. Details can be found in Appendix C.3.

● **Bounding Regret of Candidate Arm to Safe Arm.** The first sum on the LHS of (5) will be further decomposed using the *last master arm* $a_\ell$ which is the last arm to be evicted from the master arm set $\mathcal{A}_{\text{master}}$ in episode $[t_\ell, t_{\ell+1})$. Carefully using SST and STI (see Lemma 13), we further decompose $\delta_t(a_t^\sharp, \hat{a}_t)$ as:

$$\sum_{t=t_\ell}^{t_{\ell+1}-1} \delta_t(a_t^\sharp, \hat{a}_t) \leq 2 \underbrace{\sum_{t=t_\ell}^{t_{\ell+1}-1} \delta_t(a_t^\sharp, a_\ell)}_{A} + \underbrace{\sum_{t=t_\ell}^{t_{\ell+1}-1} \delta_t(a_\ell, \hat{a}_t)}_{B} + 3 \underbrace{\sum_{t=t_\ell}^{t_{\ell+1}-1} \delta_t(a_t^*, a_t^\sharp)}_{C} \qquad (6)$$

The sum C above was already bounded earlier. So, we turn our attention to B and A.

● **Bounding B.** Note that the arm $a_\ell$ by definition is never evicted by any base algorithm until the end of the episode $t_{\ell+1} - 1$. This means that at round $t \in [t_\ell, t_{\ell+1})$, the quantity $\sum_{s=t_\ell}^{t} \hat{\delta}_s(a_\ell, \hat{a}_s)$ is always kept small by the candidate arm switching criterion (4). So, by concentration (Proposition 10), we have $\sum_{s=t_\ell}^{t_{\ell+1}-1} \hat{\delta}_s(a_\ell, \hat{a}_s) \lesssim \sqrt{K(t_{\ell+1} - t_\ell)}$.

● **Bounding A** The main intuition here, similar to Appendix B.2 of Suk and Kpotufe [2022], is that well-timed replays are scheduled w.h.p. to ensure fast detection of large regret of the last master arm $a_\ell$. Key in this is the notion of a *bad segment of time*: i.e., an interval $[s_1, s_2] \subseteq [\tau_i, \tau_{i+1})$ lying inside a significant phase with last safe arm $a^\sharp$ where:

$$\sum_{t=s_1}^{s_2} \delta_t(a^\sharp, a_\ell) \gtrsim \sqrt{K \cdot (s_2 - s_1)}. \qquad (7)$$

For a fixed bad segment $[s_1, s_2]$, the idea is that a fortuitously timed replay scheduled at round $s_1$ and remaining active till round $s_2$ will evict arm $a_\ell$.

It is not immediately obvious how to carry out this argument in the dueling bandit problem since, to detect large $\sum_t \delta_t(a^\sharp, a_\ell)$, the pair of arms $a^\sharp, a_\ell$ need to both be played which, as we discussed in Section 5.1, may not occur often enough to ensure tight estimation of the gaps.

Instead, we carefully make use of SST/STI to relate $\delta_t(a^\sharp, a_\ell)$ to $\delta_t(\hat{a}_t, a_\ell)$. Note this latter quantity controls both the eviction (3) and $\hat{a}_t$ switching (4) criteria. This allows us to convert bad intervals with large $\sum_t \delta_t(a_t^\sharp, a_\ell)$ to intervals with large $\sum_t \delta_t(\hat{a}_t, a_\ell)$. Specifically, by Lemma 13, we have that (7) implies

$$2 \sum_{t=s_1}^{s_2} \delta_t(a^\sharp, \hat{a}_t) + \sum_{t=s_1}^{s_2} \delta_t(\hat{a}_t, a_\ell) + 3 \sum_{t=s_1}^{s_2} \delta_t(a_t^*, a^\sharp) \gtrsim \sqrt{K \cdot (s_2 - s_1)}. \qquad (8)$$

Then, we claim that, so long as a base algorithm Base-Alg$(s_1, m)$ is scheduled from $s_1$ running till $s_2$, we will have $\sum_{t=s_1}^{s_2} \delta_t(\hat{a}_t, a_\ell) \gtrsim \sqrt{K \cdot (s_2 - s_1)}$ which implies $a_\ell$ will be evicted. In other words, the second sum dominates the first and third sums in (8). We repeat earlier arguments to show this:

- By the definition of the last safe arm $a^\sharp$, $\sum_{t=s_1}^{s_2} \delta_t(a_t^*, a^\sharp) < \sqrt{K \cdot (s_2 - s_1)}$.
- Meanwhile, $\sum_{t=s_1}^{s_2} \delta_t(a^\sharp, \hat{a}_t) \lesssim \sqrt{K \cdot (s_2 - s_1)}$ by the candidate switching criterion (4) and because $a^\sharp$ will not be evicted before round $s_2$ lest it incurs significant regret which cannot happen by definition of $a^\sharp$.

Combining the above two points with (8), we have that $\sum_{t=s_1}^{s_2} \delta_t(\hat{a}_t, a_\ell) \gtrsim \sqrt{K \cdot (s_2 - s_1)}$, which directly aligns with our criterion (3) for evicting $a_\ell$. To summarize, a bad segment $[s_1, s_2]$ in the sense of (7) is detectable using a well-timed instance of SWIFT, which happens often enough with high probability. Concretely, we argue that not too many bad segments elapse before $a_\ell$ is evicted by a well-timed replay in the above sense and that thus the regret incurred by $a_\ell$ is bounded by the RHS of (5). The details can be found in Appendix C.5.

## 7 Conclusion

We consider the problem of switching dueling bandits where the distribution over preferences can change over time. We study a notion of significant shifts in preferences and ask whether one can achieve adaptive dynamic regret of $O(\sqrt{K\tilde{L}T})$ where $\tilde{L}$ is the number of significant shifts. We give a negative result showing that one cannot achieve such a result outside of the SST∩STI setting, and answer this question in the affirmative under the SST∩STI setting. In the future, it would be interesting to consider other notions of shifts which are weaker than the notion of significant shift, and ask whether adaptive algorithms for the Condorcet setting can be designed with respect to these notions. Buening and Saha [2022] already give a $O(K\sqrt{ST})$ bound for the Condorcet setting, where $S$ is the number of changes in 'best' arm. However, their results have a suboptimal dependence on $K$ due to reduction to "all-pairs" exploration.

## Acknowledgements

We thank Samory Kpotufe for helpful discussions. We also acknowledge computing resources from Columbia University's Shared Research Computing Facility project, which is supported by NIH Research Facility Improvement Grant 1G20RR030893-01, and associated funds from the New York State Empire State Development, Division of Science Technology and Innovation (NYSTAR) Contract C090171, both awarded April 15, 2010.

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

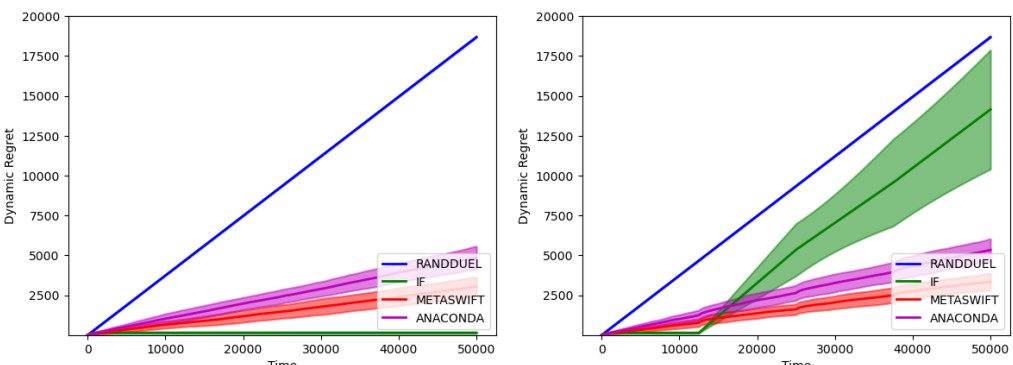

Figure 2: Plots of dynamic regret curves over time.

|              | Algorithm | Mean Regret | Standard Deviation |
|--------------|-----------|-------------|--------------------|
| $S = 0$ Changes | METASWIFT | 3048 | 600 |
|              | ANACONDA  | 4863 | 707 |
|              | IF        | 140  | 46  |
|              | RANDDUEL  | 18688 | 30 |
| $S = 4$ Changes | METASWIFT | 3346 | 531 |
|              | ANACONDA  | 5331 | 705 |
|              | IF        | 14142 | 3739 |
|              | RANDDUEL  | 18684 | 23 |

Table 1: Table of total dynamic regrets.

## A  Experiments

Code can be found at https://github.com/joesuk/nonstationary-duel.

**Synthetic Environments.**   We used a geometric BTL model where the arms are linearly ordered and the $i$-th best arm beats the $j$-th best arm with probability $\mathbb{P}(i \succ j) = \frac{2^{-i}}{2^{-j}+2^{-i}}$. At each changepoint, the ordering of arms was randomly permuted, with a total $T = 50,000$ rounds with $K = 10$ arms. Regret was computed over $N = 50$ trials of each environment with standard confidence bands shown.

**Algorithms.**   We considered four algorithms: (1) METASWIFT (Algorithm 1, (2) the ANACONDA algorithm of Buening and Saha [2022] (3) Interleaved Filtering (which we abbreviate as IF) as specified by Yue et al. [2012b], and a baseline (3) RANDDUEL which naively plays a pair of arms selected uniformly at random every round.

**Parameters.**   Parameters associated with each of the algorithms (e.g., the constants in displays (3) and (4), analogous quantities in ANACONDA, and IF's elimination threshold) were tuned using cross validation on randomly generated geometric BTL environments with number of changepoints varying from 0 to 1000. For fairness, all algorithms were given the chance to tune parameters on the same environments before testing.

The first graphic in Figure 2 shows the regret curves in a stationary environment with $S = 0$ changes. The second graphic shows the regret curves in a non-stationary environment with $S = 4$ changes. Exact mean and standard deviations on final regret are given in Table 1. These do support the theoretical message that METASWIFT performs better than the existing ANACONDA algorithm in non-stationary environments due to more efficient exploration of arms (demonstrated through $\sqrt{K}$ versus $K$ dependence in the theoretical bounds). Moreover, we also observe that the IF algorithm which is designed for stationary environments can have almost linear regret in non-stationary environments.

# B  Proof of Theorem 3

Consider the following preference matrices for some $\epsilon > 0$ (to be chosen later):

$$\mathbf{P}^+ := \begin{pmatrix} 1/2 & 1/2 + \epsilon & 1 \\ 1/2 - \epsilon & 1/2 & 1/2 + \epsilon \\ 0 & 1/2 - \epsilon & 1/2 \end{pmatrix}, \mathbf{P}^- := \begin{pmatrix} 1/2 & 1/2 - \epsilon & 0 \\ 1/2 + \epsilon & 1/2 & 1/2 - \epsilon \\ 1 & 1/2 + \epsilon & 1/2 \end{pmatrix}.$$

In environment $\mathbf{P}^+$, arm 1 is the Condorcet winner and we have $1 \succ 2 \succ 3$. In environment $\mathbf{P}^-$, arm 3 is the winner with $3 \succ 2 \succ 1$.

Consider a uniform mixture $\mathcal{U}$ of the preference matrices $\mathbf{P}^+$ and $\mathbf{P}^-$, Let $\mathcal{E}$ be a (random) sequence of $T$ environments sampled i.i.d. from $\mathcal{U}$, with $\mathbf{P}_t := (\mathcal{E})_t$ being the sampled environment at round $t$.

First, it is straightforward to verify in every such switching dueling bandit $\mathcal{E}$, arm 2 does not incur significant regret over any interval of rounds $[s_1, s_2] \subseteq [1, T]$, for $\epsilon < 1/\sqrt{T}$. Thus, every such $\mathcal{E}$ exhibits zero significant shifts.

Next, in what follows, we use $\mathbb{E}_{\mathcal{E}}[\cdot]$ to denote an expectation over both the randomness of $\mathcal{U}^{\otimes T}$ and the algorithm's feedback and decisions. If there exists a realization of $\mathcal{E}$ such that the algorithm gets expected regret at least $T/8$, then we are already done. Otherwise, we have the expected regret over the random environment $\mathcal{E}$ is bounded above by $T/8$. Next, define the *arm-pull counts* $N(T, a) := \sum_{t=1}^T \mathbf{1}\{i_t = a\} + \mathbf{1}\{j_t = a\}$ for each arm $a$. Then, we relate these arm-pull counts to the regret:

$$
\begin{aligned}
T/8 &> \sum_{t=1}^T \mathbb{E}_{\mathcal{E}}\left[\delta_t(i^*, i_t) + \delta_t(i^*, j_t)\right] \\
&\geq \frac{1}{2}\sum_{t=1}^T \mathbb{E}_{\mathcal{E}}[(\mathbf{1}\{i_t = 3\} + \mathbf{1}\{j_t = 3\}) \cdot \mathbf{1}\{(\mathcal{E})_t = \mathbf{P}^+\} \\
&\qquad + (\mathbf{1}\{i_t = 1\} + \mathbf{1}\{j_t = 1\}) \cdot \mathbf{1}\{(\mathcal{E})_t = \mathbf{P}^-\}] \\
&= \frac{1}{2}\sum_{t=1}^T \mathbb{E}_{\mathcal{E}}\left[\frac{1}{2} \cdot (\mathbf{1}\{i_t = 3\} + \mathbf{1}\{j_t = 3\} + \mathbf{1}\{i_t = 1\} + \mathbf{1}\{j_t = 1\})\right] \\
&\geq \frac{1}{4} \cdot \mathbb{E}_{\mathcal{E}}[N(T, 3) + N(T, 1)],
\end{aligned}
$$

where we use the tower law in the third inequality (note that $i_t, j_t$ are independent of $(\mathcal{E})_t$). Thus, in expectation over both the model noise and randomness of $\mathcal{E}$, arms 3 and 1 cannot be played more than $T/2$ times without causing linear regret.

Since $\sum_{a=1}^3 \mathbb{E}_{\mathcal{E}}[N(T, a)] = 2T$, we conclude that $\mathbb{E}_{\mathcal{E}}[N(T, 2)] \geq 3T/2$. We will next show that arm 2 is statistically indistinguishable from arm 3. To do so, we consider an analogous environment which is identical to $\mathcal{E}$ except the identities of arms 2 and 3 are switched. Specifically, let $\mathcal{E}'$ be a random sequence of $T$ environments sampled i.i.d. from a uniform mixture of $\mathbf{Q}^+$ and $\mathbf{Q}^-$, which are respectively $\mathbf{P}^+$ and $\mathbf{P}^-$ with switched entries for arms 2 and 3.

We next claim $\mathbb{E}_{\mathcal{E}}[N(T, 2)] = \mathbb{E}_{\mathcal{E}'}[N(T, 2)]$. Admitting this claim, it immediately follows that the algorithm has expected regret (over the randomness of $\mathcal{E}'$) at least (using an analogous chain of inequalities as above):

$$\mathbb{E}_{\mathcal{E}'}[\mathrm{DR}(T)] \geq \frac{1}{4} \cdot \mathbb{E}_{\mathcal{E}'}[N(T, 2)] \geq 3T/8.$$

In particular, there exists a realization of $\mathcal{E}'$ within the prior on environments on which the regret is at least $3T/8$.

It remains to show $\mathbb{E}_{\mathcal{E}}[N(T, 2)] = \mathbb{E}_{\mathcal{E}'}[N(T, 2)]$. This will follow from Pinsker's inequality and showing that the KL divergence between $\mathcal{E}$ and $\mathcal{E}'$ is zero.

We first observe that the dueling observations $O_t(i, j)$ at each round $t \in [T]$ are identically a $\mathrm{Ber}(1/2)$ R.V. for all pairs of arms $i, j$ in both $\mathcal{E}$ and $\mathcal{E}'$, since a uniform mixture of a $\mathrm{Ber}(1/2 + \epsilon)$ and a $\mathrm{Ber}(1/2 - \epsilon)$ is a $\mathrm{Ber}(1/2)$, while so is the uniform mixture of a $\mathrm{Ber}(1)$ and a $\mathrm{Ber}(0)$.

Then, since $N(T, 2) \leq 2T$, by Pinsker's inequality [see Gupta and Saha, 2022, proof of Lemma C.1], we have:

$$\mathbb{E}_{\mathcal{E}}[N(T, 2)] - \mathbb{E}_{\mathcal{E}'}[N(T, 2)] \leq 2T\sqrt{\frac{\mathrm{KL}(\mathcal{P}, \mathcal{P}')}{2}},$$

where $\mathcal{P}$ and $\mathcal{P}'$ are the induced distributions over the randomness $\mathcal{U}^{\otimes T}$, and the history of observations and decisions in $T$ rounds by $\mathcal{E}$ and $\mathcal{E}'$. Let $\mathcal{H}_t$ be the history of randomness, observations, and decisions till round $t$: $\mathcal{H}_t = \{(u_s, i_s, j_s, O_s(i_s, j_s))\}_{s \leq t}$ where $u_s \sim \mathrm{Ber}(1/2)$ decides whether $\mathbf{P}^+/\mathbf{Q}^+$ or $\mathbf{P}^-/\mathbf{Q}^-$ is realized at round $t$. Let $\mathcal{P}_t$ and $\mathcal{P}'_t$ denote the respective marginal distributions over the round $t$ data $(u_t, i_t, j_t, O_t(i_t, j_t))$. Then, repeatedly using chain rule for KL and then conditioning on the played arms $(i_t, j_t)$ (whose identities are fixed given $\mathcal{H}_{t-1}$) at round $t$, we get:

$$\mathrm{KL}(\mathcal{P}, \mathcal{P}') = \sum_{t=1}^{T} \mathrm{KL}(\mathcal{P}_t|\mathcal{H}_{t-1}, \mathcal{P}'_t|\mathcal{H}_{t-1}) = \sum_{t=1}^{T} \mathbb{E}_{\mathcal{H}_{t-1}}[\mathbb{E}_{i_t, j_t}[\mathrm{KL}(\mathrm{Ber}(1/2), \mathrm{Ber}(1/2))]] = 0.$$

∎

**Remark 2.** *The constructed environments $\mathbf{P}^+, \mathbf{P}^-$ in the proof of Theorem 3 satisfies SST but violates STI. A similar construction which violates SST (but satisfies STI) can also be used in the proof. Let*

$$\mathbf{P}^+ := \begin{pmatrix} 1/2 & 1/2 - \epsilon & 1/2 - \epsilon \\ 1/2 + \epsilon & 1/2 & 0 \\ 1/2 + \epsilon & 1 & 1/2 \end{pmatrix},$$

*and let $\mathbf{P}^-$ be the same preference matrix with arms $2$ and $3$ switched. Note that $3 \succ 2 \succ 1$ in $\mathbf{P}^+$ and $2 \succ 3 \succ 1$ in $\mathbf{P}^-$. Here, arm $1$ is the "safe" arm as it always has a gap of $\epsilon$ while arms $2$ and $3$ randomly alternate between being the best arm and the worst arm with a gap of $1/2$. Thus, both the STI and SST assumptions are required to get sublinear regret in mildly adversarial environments.*

Due to these observations we have the following corollaries.

**Corollary 6.** *For each horizon $T$, there exists a finite family $\mathcal{F}$ of switching dueling bandit environments with $K = 3$ that satisfies the SST condition with $\tilde{L} = 0$ significant shifts. The worst-case regret of any algorithm on an environment $\mathcal{E}$ in this family is lower bounded as*

$$\sup_{\mathcal{E} \in \mathcal{F}} \mathbb{E}_{\mathcal{E}}[\mathrm{DR}(T)] \geq T/8.$$

**Corollary 7.** *For each horizon $T$, there exists a finite family $\mathcal{F}$ of switching dueling bandit environments with $K = 3$ that satisfies the STI condition with $\tilde{L} = 0$ significant shifts. The worst-case regret of any algorithm on an environment $\mathcal{E}$ in this family is lower bounded as*

$$\sup_{\mathcal{E} \in \mathcal{F}} \mathbb{E}_{\mathcal{E}}[\mathrm{DR}(T)] \geq T/8.$$

## C  Full Proof of Theorem 4

Throughout the proof $c_1, c_2, \ldots$ will denote positive constants not depending on $T$ or any distributional parameters. First, we observe the regret bound is vacuous for $T < K$; so, assume $T \geq K$. Recall from Line 3 of Algorithm 1 that $t_\ell$ is the first round of the $\ell$-th episode. WLOG, there are $T$ total episodes and, by convention, we let $t_\ell := T + 1$ if only $\ell - 1$ episodes occurred by round $T$.

Next, we establish an elementary lemma which will help us leverage the STI and SST assumptions.

### C.1  Decomposing the Regret

**Lemma 8.** *For any three arms $b, c$, under SST$\cap$STI: $\delta_t(a_t^*, c) \leq 2 \cdot \delta_t(a_t^*, b) + \delta_t(b, c)$.*

*Proof.* If $b \succeq_t c$, this is true by STI. Otherwise, $\delta_t(a_t^*, c) \leq \delta_t(a_t^*, b) \leq \delta_t(a_t^*, b) + \delta_t(a_t^*, b) - \delta_t(c, b)$ by SST. □

Using Lemma 8 twice, we have the regret can be written as

$$\sum_{t=1}^{T} \delta_t(a_t^*, \hat{a}_t) + \delta_t(a_t^*, a_t) \leq \sum_{t=1}^{T} 6 \cdot \delta_t(a_t^*, a_t^\sharp) + 3 \cdot \delta_t(a_t^\sharp, \hat{a}_t) + \delta_t(\hat{a}_t, a_t).$$

Following the discussion of Section 6, it remains to bound $\sum_{t=1}^{T} \delta_t(a_t^\sharp, \hat{a}_t)$ and $\sum_{t=1}^{T} \delta_t(\hat{a}_t, a_t)$ in expectation. For this, we need to relate our estimators $\hat{\delta}_t(\hat{a}_t, a)$ to the true gaps $\delta_t(\hat{a}_t, a)$.

## C.2  Relating Estimated Gaps to Regret

We first recall a version of Freedman's martingale concentration inequality, identical to the one used in Suk and Kpotufe [2022], Buening and Saha [2022].

**Lemma 9** (Theorem 1 of Beygelzimer et al. [2011]). *Let $X_1, \ldots, X_n \in \mathbb{R}$ be a martingale difference sequence with respect to some filtration $\{\mathcal{F}_0, \mathcal{F}_1, \ldots\}$. Assume for all $t$ that $X_t \leq R$ a.s. and that $\sum_{i=1}^{n} \mathbb{E}[X_i^2 | \mathcal{F}_{i-1}] \leq V_n$ a.s. for some constant $V_n$ only depending on $n$. Then for any $\delta \in (0, 1)$, with probability at least $1 - \delta$, we have:*

$$\sum_{i=1}^{n} X_i \leq (e-1)\left(\sqrt{V_n \log(1/\delta)} + R \log(1/\delta)\right).$$

We next apply Lemma 9 to bound the estimation error of our estimates $\hat{\delta}_t(\hat{a}_t, a)$, found in (2).

**Proposition 10.** *Let $\mathcal{E}_1$ be the event that for all rounds $s_1 < s_2$ and all arms $a \in [K]$:*

$$\left| \sum_{t=s_1}^{s_2} \hat{\delta}_t(\hat{a}_t, a) - \sum_{t=s_1}^{s_2} \mathbb{E}\left[\hat{\delta}_t(\hat{a}_t, a) \mid \mathcal{F}_{t-1}\right] \right| \leq c_1 \log(T)\left(\sqrt{K(s_2 - s_1)} + K\right), \qquad (9)$$

*for an appropriately large constant $c_1$, and where $\mathcal{F} := \{\mathcal{F}_t\}_{t=1}^{T}$ is the canonical filtration generated by observations and randomness of elapsed rounds. Then, $\mathcal{E}_1$ occurs with probability at least $1 - 1/T^2$.*

*Proof.* The random variable $\hat{\delta}_t(\hat{a}_t, a) - \mathbb{E}[\hat{\delta}_t(\hat{a}_t, a)|\mathcal{F}_{t-1}]$ is a martingale difference bounded above by $K$ for all rounds $t$ and all arms $a, a'$. Note here that the identity of the candidate arm $\hat{a}_t$ is fixed conditional on the observations of the previous rounds $\mathcal{F}_{t-1}$. The variance of this difference is:

$$\sum_{t=s_1}^{s_2} \mathbb{E}[\hat{\delta}_t^2(\hat{a}_t, a) \mid \mathcal{F}_{t-1}] \leq \sum_{t=s_1}^{s_2} |\mathcal{A}_t|^2 \mathbb{E}[\mathbf{1}\{j_t = a\}|\mathcal{F}_{t-1}]$$

$$\leq \sum_{t=s_1}^{s_2} |\mathcal{A}_t|^2 \cdot \frac{1}{|\mathcal{A}_t|}$$

$$\leq K \cdot (s_2 - s_1 + 1).$$

$$\leq 2K \cdot (s_2 - s_1)$$

Then, the result follows from Lemma 9 and taking union bounds over arms $a$ and rounds $s_1, s_2$. $\qquad \square$

Since the contribution to the expected regret is small outside of the high-probability good event $\mathcal{E}_1$, going forward we will assume as necessary that (9) holds for all arms $a \in [K]$ and rounds $s_1, s_2$. The next result asserts that episodes roughly correspond to significant shifts in the sense that a restart (Line 8 of Algorithm 1) occurs only if a significant shift has been detected.

**Lemma 11.** *On event $\mathcal{E}_1$, for each episode $[t_\ell, t_{\ell+1})$ with $t_{\ell+1} \leq T$ (i.e., an episode which concludes with a restart), there exists a significant shift $\tau_i \in [t_\ell, t_{\ell+1})$.*

*Proof.* We have that

$$\mathbb{E}[\hat{\delta}_t(\hat{a}_t, a)|\mathcal{F}_{t-1}] = \begin{cases} \delta_t(\hat{a}_t, a) & a \in \mathcal{A}_t \\ -1/2 & a \notin \mathcal{A}_t \end{cases}.$$

Thus, by concentration (Proposition 10) and the eviction criteria (3) with large enough constant $C > 0$, we have that an arm $a$ being evicted over interval $[s_1, s_2]$ implies $\sum_{t=s_1}^{s_2} \delta_t(\hat{a}_t, a) > \sqrt{K \cdot (s_2 - s_1)}$. By the SST condition, this means that

$$\sum_{t=s_1}^{s_2} \delta_t(a_t^*, a) \geq \sum_{t=s_1}^{s_2} \delta_t(\hat{a}_t, a) > \sqrt{K \cdot (s_2 - s_1)}.$$

This means, over the course of episode $[t_\ell, t_{\ell+1})$, every arm $a \in [K]$ incurs significant regret meaning a significant shift must take place between rounds $t_\ell$ and $t_{\ell+1} - 1$. $\qquad \square$

Following the outline of Section 6, we now turn our attention to bounding the regrets $\delta_t(a_t^\sharp, \hat{a}_t)$ and $\delta_t(\hat{a}_t, a_t)$ over a single episode $[t_\ell, t_{\ell+1})$.

## C.3  Bounding $\mathbb{E}[\sum_{t=t_\ell}^{t_{\ell+1}-1} \delta_t(\hat{a}_t, a_t)]$: Regret of Active Arms to Candidate Arm

We first decompose the total sum of regrets $\mathbb{E}[\sum_{t=1}^{T} \delta_t(\hat{a}_t, a_t)]$ based on which arm $a_t$ chooses within the active set $\mathcal{A}_t$. Using tower law, we have

$$\mathbb{E}\left[\sum_{t=1}^{T} \delta_t(\hat{a}_t, a_t)\right] = \sum_{t=1}^{T} \mathbb{E}[\mathbb{E}[\delta_t(\hat{a}_t, a_t) \mid \mathcal{F}_{t-1}]] = \mathbb{E}\left[\sum_{t=1}^{T} \sum_{a \in \mathcal{A}_t} \frac{\delta_t(\hat{a}_t, a)}{|\mathcal{A}_t|}\right].$$

Splitting the above RHS back along episodes, we obtain the sum $\mathbb{E}[\sum_{t=t_\ell}^{t_{\ell+1}-1} \sum_{a \in \mathcal{A}_t} \delta_t(\hat{a}_t, a)/|\mathcal{A}_t|]$.

Next, we condition on the good event $\mathcal{E}_1$ on which the concentration bounds of Proposition 10 hold. Additionally, we divide up the rounds $t$ into those before arm $a$ is evicted from $\mathcal{A}_{\text{master}}$ and those after. Suppose arm $a$ is evicted from $\mathcal{A}_{\text{master}}$ at round $t_\ell^a \in [t_\ell, t_{\ell+1})$. In particular, this means arm $a \in \mathcal{A}_t$ for all $t \in [t_\ell, t_\ell^a)$. Thus, it suffices to bound:

$$\mathbb{E}\left[\mathbf{1}\{\mathcal{E}_1\} \cdot \left(\sum_{a=1}^{K} \sum_{t=t_\ell}^{t_\ell^a-1} \frac{\delta_t(\hat{a}_t, a)}{|\mathcal{A}_t|} + \sum_{a=1}^{K} \sum_{t=t_\ell^a}^{t_{\ell+1}-1} \frac{\delta_t(\hat{a}_t, a)}{|\mathcal{A}_t|} \cdot \mathbf{1}\{a \in \mathcal{A}_t\}\right)\right]. \tag{10}$$

Suppose WLOG that $t_\ell^1 \leq t_\ell^2 \leq \cdots \leq t_\ell^K$. Then, for each round $t < t_\ell^a$ all arms $a' \geq a$ are retained in $\mathcal{A}_{\text{master}}$ and thus retained in the candidate arm set $\mathcal{A}_t$. Thus, $|\mathcal{A}_t| \geq K + 1 - a$ for all $t \leq t_\ell^a$.

Then, the first double sum in (10) can be bounded by combining our eviction criterion (3) with our concentration bounds Proposition 10. Since arm $a$ is not evicted from $\mathcal{A}_t$ till round $t_\ell^a$, on event $\mathcal{E}_1$ we have for some $c_2 > 0$:

$$\sum_{t=t_\ell}^{t_\ell^a-1} \delta_t(\hat{a}_t, a) = \sum_{t=t_\ell}^{t_\ell^a-1} \mathbb{E}[\hat{\delta}_t(\hat{a}_t, a) \mid \mathcal{F}_{t-1}] \leq c_2 \log(T)\sqrt{K(t_\ell^a - t_\ell) \vee K^2}$$

Then, using the fact that $|\mathcal{A}_t| \geq K + 1 - a$ for all $t \in [t_\ell, t_\ell^a)$, we have:

$$\sum_{t=t_\ell}^{t_\ell^a-1} \frac{\delta_t(\hat{a}_t, a)}{|\mathcal{A}_t|} \leq \frac{c_2 \log(T)\sqrt{K(t_\ell^a - t_\ell) \vee K^2}}{K + 1 - a}.$$

Then, summing the above R.H.S. over all arms $a$, we have on event $\mathcal{E}_1$:

$$\sum_{a=1}^{K} \sum_{t=t_\ell}^{t_\ell^a-1} \frac{\delta_t(\hat{a}_t, a)}{|\mathcal{A}_t|} \leq c_2 \log(K) \log(T)\sqrt{K(t_{\ell+1} - t_\ell) \vee K^2}.$$

Next, we handle the second double sum in (10). We first observe that if arm $a$ is played after round $t_\ell^a$, then it must due to a scheduled replay. The difficulty here is that replays may interrupt each other and so care must be taken in managing the relative regret contribution $\sum_t \delta_t(\hat{a}_t, a)$ (which may be negative if $a \prec \hat{a}_t$) of different overlapping replays.

Fixing an arm $a$, our strategy is to partition the rounds when $a$ is played by a replay after round $t_\ell^a$ according to which replay is active and not accounted for by another replay. This involves carefully designating a subclass of replays whose durations while playing $a$ span all the rounds where $a$ is played after $t_\ell^a$. Then, we cover the times when $a$ is played by a collection of intervals corresponding to the schedules of this subclass of replays, on each of which we can employ the eviction criterion (3) and concentration like before.

For this purpose, we define the following terminology (which is all w.r.t. a fixed arm $a$):

**Definition 4.**

   (i) *For each scheduled and activated* Base-Alg$(s, m)$*, let the round* $M(s, m)$ *be the minimum of two quantities: (a) the last round in* $[s, s+m]$ *when arm* $a$ *is retained by* Base-Alg$(s, m)$ *and all of its children, and (b) the last round that* Base-Alg$(s, m)$ *is active and not permanently interrupted. Call the interval* $[s, M(s, m)]$ *the* **active interval** *of* Base-Alg$(s, m)$*.*

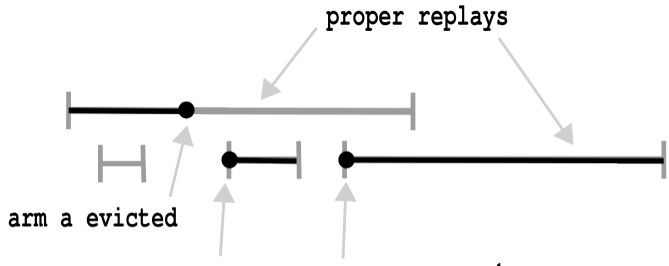

proper replays

arm a evicted

subproper replays reintroduce arm a to $\mathcal{A}_t$

Figure 3: Shown are replay scheduled durations (in gray) with dots marking when arm $a$ is reintroduced to $\mathcal{A}_t$. Black segments indicate the period $[s, M(s, m)]$ for proper and subproper replays. Note that the rounds where $a \in \mathcal{A}_t$ in the left unlabeled replay's duration are accounted for by the larger proper replay.

(ii) *Call a replay* Base-Alg$(s, m)$ **proper** *if there is no other scheduled replay* Base-Alg$(s', m')$ *such that* $[s, s + m] \subset (s', s' + m')$ *where* Base-Alg$(s', m')$ *will become active again after round* $s + m$. *In other words, a proper replay is not scheduled inside the scheduled range of rounds of another replay. Let* PROPER$(t_\ell, t_{\ell+1})$ *be the set of proper replays scheduled to start before round* $t_{\ell+1}$.

(iii) *Call a scheduled replay* Base-Alg$(s, m)$ **subproper** *if it is non-proper and if each of its ancestor replays (i.e., previously scheduled replays whose durations have not concluded)* Base-Alg$(s', m')$ *satisfies* $M(s', m') < s$. *In other words, a subproper replay either permanently interrupts its parent or does not, but is scheduled after its parent (and all its ancestors) stops playing arm* $a$. *Let* SUBPROPER$(t_\ell, t_{\ell+1})$ *be the set of all subproper replays scheduled before round* $t_{\ell+1}$.

Equipped with this language, we now show some basic claims which essentially reduce analyzing the complicated hierarchy of replays to analyzing the active intervals of replays in PROPER$(t_\ell, t_{\ell+1}) \cup$ SUBPROPER$(t_\ell, t_{\ell+1})$.

**Proposition 12.** *The active intervals*

$$\{[s, M(s, m)] : \text{Base-Alg}(s, m) \in \text{PROPER}(t_\ell, t_{\ell+1}) \cup \text{SUBPROPER}(t_\ell, t_{\ell+1})\},$$

*are mutually disjoint.*

*Proof.* Clearly, the classes of replays PROPER$(t_\ell, t_{\ell+1})$ and SUBPROPER$(t_\ell, t_{\ell+1})$ are disjoint. Next, we show the respective active intervals $[s, M(s, m)]$ and $[s', M(s', m')]$ of any two Base-Alg$(s, m)$ and Base-Alg$(s', m') \in \text{PROPER}(t_\ell, t_{\ell+1}) \cup \text{SUBPROPER}(t_\ell, t_{\ell+1})$ are disjoint.

1. Proper replay vs. subproper replay: a subproper replay can only be scheduled after the round $M(s, m)$ of the most recent proper replay Base-Alg$(s, m)$ (which is necessarily an ancestor). Thus, the active intervals of proper replays and subproper replays are disjoint.

2. Two distinct proper replays: two such replays can only permanently interrupt each other, and since $M(s, m)$ always occurs before the permanent interruption of Base-Alg$(s, m)$, we have the active intervals of two such replays are disjoint.

3. Two distinct subproper replays: consider two non-proper replays Base-Alg$(s, m)$, Base-Alg$(s', m') \in \text{SUBPROPER}(t_\ell, t_{\ell+1})$ with $s' > s$. The only way their active intervals intersect is if Base-Alg$(s, m)$ is an ancestor of Base-Alg$(s', m')$. Then, if Base-Alg$(s', m')$ is subproper, we must have $s' > M(s, m)$, which means that $[s', M(s', m')]$ and $[s, M(s, m)]$ are disjoint.

$\square$

Next, we claim that the active intervals $[s, M(s, m)]$ for Base-Alg$(s, m) \in \text{PROPER}(t_\ell, t_{\ell+1}) \cup$ SUBPROPER$(t_\ell, t_{\ell+1})$ contain all the rounds where $a$ is played after being evicted from $\mathcal{A}_{\text{master}}$. To show this, we first observe that for each round $t$ when a replay is active, there is a unique proper

replay associated to $t$, namely the proper replay scheduled most recently. Next, note that any round $t > t_\ell^a$ where arm $a \in \mathcal{A}_t$ must belong to the active interval $[s, M(s,m)]$ of the unique proper replay Base-Alg$(s,m)$ associated to round $t$, or else satisfies $t > M(s,m)$ in which case a unique subproper replay Base-Alg$(s',m') \in$ SubProper$(t_\ell, t_{\ell+1})$ was active and not yet permanently interrupted by round $t$. Thus, it must be the case that $t \in [s', M(s',m')]$.

At the same time, every round $t \in [s, M(s,m)]$ for a proper or subproper Base-Alg$(s,m)$ is clearly a round where $a \in \mathcal{A}_t$ and no such round is accounted for twice by Proposition 12. Thus,

$$\{t \in (t_\ell^a, t_{\ell+1}) : a \in \mathcal{A}_t\} = \bigsqcup_{\text{Base-Alg}(s,m) \in \text{Proper}(t_\ell, t_{\ell+1}) \cup \text{SubProper}(t_\ell, t_{\ell+1})} [s, M(s,m)].$$

Then, we can rewrite the second double sum in (10) as:

$$\sum_{a=1}^{K} \sum_{\text{Base-Alg}(s,m) \in \text{Proper}(t_\ell, t_{\ell+1}) \cup \text{SubProper}(t_\ell, t_{\ell+1})} \mathbf{1}\{B_{s,m} = 1\} \sum_{t=s \vee t_\ell^a}^{M(s,m)} \frac{\delta_t(\hat{a}_t, a)}{|\mathcal{A}_t|}.$$

Recall in the above that the Bernoulli $B_{s,m}$ (see Line 6 of Algorithm 1) decides whether Base-Alg$(s,m)$ is scheduled.

Further bounding the sum over $t$ above by its positive part, we can expand the sum over Base-Alg$(s,m) \in$ Proper$(t_\ell, t_{\ell+1}) \cup$ SubProper$(t_\ell, t_{\ell+1})$ to be over all Base-Alg$(s,m)$, or obtain:

$$\sum_{a=1}^{K} \sum_{\text{Base-Alg}(s,m)} \mathbf{1}\{B_{s,m} = 1\} \left( \sum_{t=s \vee t_\ell^a}^{M(s,m)} \frac{\delta_t(\hat{a}_t, a)}{|\mathcal{A}_t|} \cdot \mathbf{1}\{a \in \mathcal{A}_t\} \right)_+, \tag{11}$$

where the sum is over all replays Base-Alg$(s,m)$, i.e. $s \in \{t_\ell + 1, \ldots, t_{\ell+1} - 1\}$ and $m \in \{2, 4, \ldots, 2^{\lceil \log(T) \rceil}\}$. It then remains to bound the contributed relative regret of each Base-Alg$(s,m)$ in the interval $[s \vee t_\ell^a, M(s,m)]$, which will follow similarly to the previous steps. Fix $s, m$ and suppose $t_\ell^a + 1 \le M(s,m)$ since otherwise Base-Alg$(s,m)$ contributes no regret in (11).

Then, following similar reasoning as before, i.e. combining our concentration bound (9) with the eviction criterion (3), we have for a fixed arm $a$:

$$\sum_{t=s \vee t_\ell^a}^{M(s,m)} \frac{\delta_t(\hat{a}_t, a)}{|\mathcal{A}_t|} \le \frac{c_2 \log(T) \sqrt{Km \vee K^2}}{\min_{t \in [s, M(s,m)]} |\mathcal{A}_t|},$$

Plugging this into (11) and switching the ordering of the outer double sum, we obtain (now for clarity overloading the notation $M(s,m,a)$ to also depend on the reference arm $a$):

$$\sum_{\text{Base-Alg}(s,m)} \mathbf{1}\{B_{s,m} = 1\} \cdot c_2 \log(T) \sqrt{Km \vee K^2} \sum_{a=1}^{K} \frac{1}{\min_{t \in [s, M(s,m.a)]} |\mathcal{A}_t|}.$$

We claim the above innermost sum over $a$ is at most $\log(K)$. For a fixed Base-Alg$(s,m)$, if $a_k$ is the $k$-th arm in $[K]$ to be evicted by Base-Alg$(s,m)$ or any of its children, then $\min_{t \in [s, M(s,m,a_k)]} |\mathcal{A}_t| \ge K + 1 - k$. Thus, our claim follows follows from $\sum_{k=1}^{K} \frac{1}{K+1-k} \le \log(K)$.

Let $R(m) := c_2 \log(K) \log(T) \sqrt{Km \vee K^2}$ which is the bound we've obtained so far on the relative regret for a single Base-Alg$(s,m)$. Then, plugging $R(m)$ into (11) gives:

$$\mathbb{E}\left[ \mathbf{1}\{\mathcal{E}_1\} \sum_{a=1}^{K} \sum_{t=t_\ell^a}^{t_{\ell+1}-1} \frac{\delta_t(\hat{a}_t, a)}{|\mathcal{A}_t|} \cdot \mathbf{1}\{a \in \mathcal{A}_t\} \right] \le \mathbb{E}_{t_\ell}\left[ \mathbb{E}\left[ \sum_{\text{Base-Alg}(s,m)} \mathbf{1}\{B_{s,m} = 1\} \cdot R(m) \mid t_\ell \right] \right]$$

$$= \mathbb{E}_{t_\ell}\left[ \sum_{s=t_\ell}^{T} \sum_{m} \mathbb{E}[\mathbf{1}\{B_{s,m} = 1\} \cdot \mathbf{1}\{s < t_{\ell+1}\} \mid t_\ell] \cdot R(m) \right].$$

Next, we observe that $B_{s,m}$ and $\mathbf{1}\{s < t_{\ell+1}\}$ are independent conditional on $t_\ell$ since $\mathbf{1}\{s < t_{\ell+1}\}$ only depends on the scheduling and observations of base algorithms scheduled before round $s$. Thus,

recalling that $\mathbb{P}(B_{s,m} = 1) = 1/\sqrt{m \cdot (s - t_\ell)}$,

$$\mathbb{E}[\mathbf{1}\{B_{s,m} = 1\} \cdot \mathbf{1}\{s < t_{\ell+1}\} \mid t_\ell] = \mathbb{E}[\mathbf{1}\{B_{s,m} = 1\} \mid t_\ell] \cdot \mathbb{E}[\mathbf{1}\{s < t_{\ell+1}\} \mid t_\ell]$$
$$= \frac{1}{\sqrt{m \cdot (s - t_\ell)}} \cdot \mathbb{E}[\mathbf{1}\{s < t_{\ell+1}\} \mid t_\ell].$$

Plugging this into our expectation from before and unconditioning, we obtain:

$$\mathbb{E}\left[ \sum_{s=t_\ell+1}^{t_{\ell+1}-1} \sum_{n=1}^{\lceil \log(T) \rceil} \frac{1}{\sqrt{2^n \cdot (s - t_\ell)}} \cdot R(2^n) \right] \leq c_3 \log^3(T) \mathbb{E}_{t_\ell, t_{\ell+1}}\left[ \sqrt{K(t_{\ell+1} - t_\ell) \vee K^2} \right]. \quad (12)$$

Then, it suffices to bound $\sqrt{K(t_{\ell+1} - t_\ell) \vee K^2}$. First, we claim that every phase $[\tau_i, \tau_{i+1})$ is length at least $K/4$. Observe by our notion of significant regret, that an arm $a$ incurring significant regret on the interval $[s_1, s_2]$ means

$$\sum_{t=s_1}^{s_2} \delta_t(a_t^*, a) \geq \sqrt{K \cdot (s_2 - s_1)} \implies 2 \cdot (s_2 - s_1) \geq \sqrt{K \cdot (s_2 - s_1)} \implies s_2 - s_1 \geq K/4.$$

Thus, each significant phase (Definition 3) must be at least $K/4$ rounds long meaning $\tau_{i+1} - \tau_i = (\tau_{i+1} - \tau_i) \vee K/4$. This will allow us to remove the "$\vee K^2$" in (12). In particular, since the episode length $t_{\ell+1} - t_\ell$ in (12) can be upper bounded by the combined length of all significant phases $[\tau_i, \tau_{i+1})$ interesecting episode $[t_\ell, t_{\ell+1})$, (12) gives us the desired bound.

## C.4   Bounding $\mathbb{E}[\sum_{t=t_\ell}^{t_{\ell+1}-1} \delta_t(a_t^\sharp, \hat{a}_t)]$: Regret of Candidate Arm to Safe Arm

We first invoke an elementary lemma based on SST and STI to further help us decompose the regret.

**Lemma 13.** *For any three arms $a, b, c$, under* SST∩STI*:*

$$\delta_t(a, c) \leq 2 \cdot \delta_t(a, b) + \delta_t(b, c) + 3 \cdot \delta_t(a_t^*, a),$$

*where $a_t^*$ is the winner arm.*

*Proof.* We handle all the different orderings:

(a) $a \succ_t b, c$: this already follows from Lemma 8 since then $\delta_t(a, c) \leq 2 \cdot \delta_t(a, b) + \delta_t(b, c)$.

(b) $c \succ_t a \succ_t b$: $\delta_t(a, c) \leq 0 \leq \delta_t(a, b)$ and $\delta_t(a^*, b) \geq \delta_t(c, b)$ by SST. Summing these together gives the result.

(c) $b \succ_t a \succ_t c$: $\delta_t(a, c) \leq \delta_t(b, c)$ and $\delta_t(a_t^*, a) \geq \delta_t(b, a)$ by SST. Summing these together gives the result.

(d) $b, c \succ_t a$: $\delta_t(a^*, a)$ dominates the first two terms on the desired inequality's RHS.

$\square$

Then, using Lemma 13, we further decompose the regret about the *last master arm* $a_\ell$ defined in Section 4, which is the last arm to be evicted from $\mathcal{A}_{\text{master}}$ in episode $[t_\ell, t_{\ell+1})$. We have

$$\sum_{t=t_\ell}^{t_{\ell+1}-1} \delta_t(a_t^\sharp, \hat{a}_t) \leq 2 \sum_{t=t_\ell}^{t_{\ell+1}-1} \delta_t(a_t^\sharp, a_\ell) + \sum_{t=t_\ell}^{t_{\ell+1}-1} \delta_t(a_\ell, \hat{a}_t) + 3 \sum_{t=t_\ell}^{t_{\ell+1}-1} \delta_t(a_t^*, a_t^\sharp). \quad (13)$$

As said earlier, the sum $\sum_{t=t_\ell}^{t_{\ell+1}-1} \delta_t(a_t^*, a_t^\sharp)$ is of the right order. Meanwhile, the sum $\sum_{t=t_\ell}^{t_{\ell+1}-1} \delta_t(a_\ell, \hat{a}_t)$ is bounded using our candidate arm switching criterion (4). If $\hat{a}_t = a_\ell$ for every round $t \in [t_\ell, t_{\ell+1})$ we are already done. Otherwise, let $m_\ell$ be the last round that $a_\ell$ is not the candidate arm $\hat{a}_t$. Then, we must have that since arm $a_\ell$ is not evicted until round $t_{\ell+1} - 1$:

$$\sum_{t=t_\ell}^{t_{\ell+1}-1} \hat{\delta}_t(a_\ell, \hat{a}_t) = \sum_{t=t_\ell}^{m_\ell-1} \hat{\delta}_t(a_\ell, \hat{a}_t) \leq C \log(T) \sqrt{K \cdot (m_\ell - t_\ell) \vee K^2}$$

Then, by concentration (Proposition 10) and the fact from earlier that each phase $[\tau_i, \tau_{i+1})$ is at least $K/4$ rounds (so that "$\vee K^2$" can be removed in the above), we have that $\sum_{t=t_\ell}^{t_{\ell+1}-1} \delta_t(a_\ell, \hat{a}_t)$ is of the right order.

Then, turning back to (13), it remains to bound the regret of $a_\ell$ to $a_t^\sharp$ over the episode $[t_\ell, t_{\ell+1})$.

## C.5   Bounding $\mathbb{E}[\sum_{t=t_\ell}^{t_{\ell+1}-1} \delta_t(a_t^\sharp, a_\ell)]$: Regret of Last Master Arm to Safe Arm

First, following the outline of Section 4, we recall the definition of the *last safe arm* $a_t^\sharp$ at round $t$ which is the last arm to incur significant regret in the unique phase $[\tau_i, \tau_{i+1})$ containing round $t$.

We next formally define a bad segment, alluded to in Section 4. In what follows, bad segments will be defined with respect to a fixed arm $a$ and conditional on the episode start time $t_\ell$. We will then show that, with respect to any arm $a$, not too many bad segments will elapse before $a$ is evicted from $\mathcal{A}_{\text{master}}$. In particular, this will hold for $a = a_\ell$ which will ultimately be used to bound $\delta_t(a_t^\sharp, a_\ell)$ across the episode $[t_\ell, t_{\ell+1})$.

**Definition 5.** *Fix the episode start time $t_\ell$, and let $[\tau_i, \tau_{i+1})$ be any phase intersecting $[t_\ell, T)$. For any arm $a$, define rounds $s_{i,0}(a), s_{i,1}(a), s_{i,2}(a) \ldots \in [t_\ell \vee \tau_i, \tau_{i+1})$ recursively as follows: let $s_{i,0}(a) := t_\ell \vee \tau_i$ and define $s_{i,j}(a)$ as the smallest round in $(s_{i,j-1}(a), \tau_{i+1})$ such that arm $a$ satisfies for some fixed $c_4 > 0$:*

$$\sum_{t=s_{i,j-1}(a)}^{s_{i,j}(a)} \delta_t(a_t^\sharp, a) \geq c_4 \log(T) \sqrt{K \cdot (s_{i,j}(a) - s_{i,j-1}(a))}, \tag{14}$$

*if such a round $s_{i,j}(a)$ exists. Otherwise, we let the $s_{i,j}(a) := \tau_{i+1} - 1$. We refer to any interval $[s_{i,j-1}(a), s_{i,j}(a))$ as a **critical segment**, and as a **bad segment** (w.r.t. arm $a$) if (14) above holds.*

Note that the above definition only depends on the arm $a$ and the episode start time $t_\ell$ and that, conditional on these variables, they are fixed in the environment. Observe also that the arm $a_t^\sharp$ is fixed within any critical segment $[s_{i,j-1}(a), s_{i,j}(a)) \subseteq [\tau_i, \tau_{i+1})$ since a significant shift does not occur inside $[\tau_i, \tau_{i+1})$.

Now relating this notion of a bad segment to our goal of bounding regret, a given bad segment $[s_{i,j}(a), s_{i,j}(a))$ only contributes order $\sqrt{K \cdot (s_{i,j}(a) - s_{i,j-1}(a))}$ to the regret of $a$ to $a_t^\sharp$. At the same time, we claim that a well-timed replay (see Definition 6 below) running from $s_{i,j-1}(a)$ to $s_{i,j}(a)$ is capable of evicting arm $a$. This in turn allows us to reduce the regret bounding to studying the number and lengths of bad segments which elapse before one is detected by such a replay.

We first define such a well-timed and *perfect* replay.

**Definition 6.** *Let $\tilde{s}_{i,j}(a) := \lceil \frac{s_{i,j}(a) + s_{i,j+1}(a)}{2} \rceil$ denote the approximate midpoint of $[s_{i,j}(a), s_{i,j+1}(a))$. Given a bad segment $[s_{i,j}(a), s_{i,j+1}(a))$, define a **perfect replay** w.r.t. $[s_{i,j}(a), s_{i,j+1}(a))$ as a call of $\text{Base-Alg}(t_{\text{start}}, m)$ where $t_{\text{start}} \in [s_{i,j}(a), \tilde{s}_{i,j}(a)]$ and $m \geq s_{i,j+1}(a) - s_{i,j}(a)$*

Next, we analyze the behavior of a perfect replay on the bad segment $[s_{i,j}(a), s_{i,j+1}(a))$. Going forward, we will use the simpler notation $a_i^\sharp$ to denote the last safe arm of a phase $[\tau_i, \tau_{i+1})$, known in context.

**Proposition 14.** *Suppose the good event $\mathcal{E}_1$ holds (cf. Proposition 10). Let $[s_{i,j}(a), s_{i,j+1}(a))$ be a bad segment with respect to arm $a$. Fix an integer $m \geq s_{i,j+1}(a) - s_{i,j}(a)$. Then, if a perfect replay with respect to $[s_{i,j}(a), s_{i,j+1}(a))$ is scheduled, arm $a$ will be evicted from $\mathcal{A}_{\text{master}}$ by round $s_{i,j+1}(a)$.*

*Proof.* Suppose event $\mathcal{E}_1$ (i.e., our concentration bound (9)) holds. We first observe that by elementary calculations and the definition of the rounds $s_{i,j}(a)$, we have (in an identical fashion to Lemma 4 of Suk and Kpotufe [2022]):

$$\sum_{t=\tilde{s}_{i,j}(a)}^{s_{i,j+1}(a)} \delta_t(a_i^\sharp, a) \geq \frac{c_4}{4} \log(T) \sqrt{K (s_{i,j+1}(a) - \tilde{s}_{i,j}(a))}, \tag{15}$$

where $\tilde{s}_{i,j}(a)$ is the midpoint of $[s_{i,j}(a), s_{i,j+1}(a))$ as defined in Definition 6. The above will come in handy in showing arm $a$ is evicted over the second half of the bad segment $[\tilde{s}_{i,j}(a), s_{i,j+1}(a)]$.

Next, following the intuition given in Section 4, in order to relate $\delta_t(a_i^\sharp, a)$ to $\delta_t(\hat{a}_t, a)$, we again use SST and STI via Lemma 13 on inequality (15):

$$\sum_{t=\tilde{s}_{i,j}(a)}^{s_{i,j+1}(a)} 2 \cdot \delta_t(a_i^\sharp, \hat{a}_t) + \delta_t(\hat{a}_t, a) + 3 \cdot \delta_t(a_t^*, a_i^\sharp) \geq \frac{c_4}{4} \log(T) \sqrt{K \left(s_{i,j+1}(a) - \tilde{s}_{i,j}(a)\right)}. \quad (16)$$

We next show that $\sum_{t=\tilde{s}_{i,j}(a)}^{s_{i,j+1}(a)} \delta_t(a_i^\sharp, \hat{a}_t)$ and $\sum_{t=\tilde{s}_{i,j}(a)}^{s_{i,j+1}(a)} \delta_t(a_t^*, a_i^\sharp)$ on the above LHS are small.

First, it is clear that any perfect replay $\mathsf{Base\text{-}Alg}(t_{\text{start}}, m)$ will not evict $a_i^\sharp$ since otherwise it incurs significant regret within phase $[\tau_i, \tau_{i+1})$ (see the earlier Lemma 11). At the same time, by the candidate arm switching criterion (4) and concentration:

$$\sum_{t=\tilde{s}_{i,j}(a)}^{s_{i,j+1}(a)} \delta_t(a_i^\sharp, \hat{a}_t) \leq c_5 \log(T) \sqrt{K \left(s_{i,j+1}(a) - \tilde{s}_{i,j}(a)\right)}.$$

Meanwhile, by the definition of significant regret (Definition 3),

$$\sum_{t=\tilde{s}_{i,j}(a)}^{s_{i,j+1}(a)} \delta_t(a_t^*, a_i^\sharp) \geq \sqrt{K \left(s_{i,j+1}(a) - \tilde{s}_{i,j}(a)\right)}.$$

Thus, for sufficiently large $c_4 > 0$ in the definition of bad segments (Definition 5), we have that the above two inequalities can be combined with (16) to yield:

$$\sum_{t=\tilde{s}_{i,j}(a)}^{s_{i,j+1}(a)} \delta_t(\hat{a}_t, a) \geq \sqrt{K \left(s_{i,j+1}(a) - \tilde{s}_{i,j}(a)\right)}.$$

If arm $a$ is evicted from $\mathcal{A}_{\text{master}}$ before round $s_{i,j+1}(a)$, then we are already done. Otherwise, using the fact that $\mathbb{E}[\hat{\delta}_t(\hat{a}_t, a) | \mathcal{F}_{t-1}] = \delta_t(\hat{a}_t, a)$ for any round $t \in [\tilde{s}_{i,j}(a), s_{i,j+1}(a)]$ with $a \in \mathcal{A}_t$, we have that arm $a$ will be evicted at round $s_{i,j+1}(a)$ using the above inequality and concentration. $\square$

It remains to show that, for any arm $a$, a perfect replay is scheduled w.h.p. before too much regret is incurred on the elapsed bad segments w.r.t. $a$. In particular, this will hold for the last master arm $a_\ell$, allowing us to bound the remaining term $\mathbb{E}[\sum_{t=t_\ell}^{t_{\ell+1}-1} \delta_t(a_t^\sharp, a_\ell)]$. The argument will be identical to that of Appendix B.2 of Suk and Kpotufe [2022].

First, fix an arm $a$ and an episode start time $t_\ell$. Then, define the *bad round* $s(a) > t_\ell$ as follows:

**Definition 7.** *(bad round) For a fixed round $t_\ell$ and arm $a$, the **bad round** $s(a) > t_\ell$ is defined as the smallest round which satisfies, for some fixed $c_6 > 0$:*

$$\sum_{(i,j)} \sqrt{s_{i,j+1}(a) - s_{i,j}(a)} > c_6 \log(T) \sqrt{s(a) - t_\ell}, \quad (17)$$

*where the above sum is over all pairs of indices $(i,j) \in \mathbb{N} \times \mathbb{N}$ such that $[s_{i,j}(a), s_{i,j+1}(a))$ is a bad segment with $s_{i,j+1}(a) < s(a)$.*

Our goal is then to then to show that arm $a$ is evicted by some perfect replay scheduled within episode $[t_\ell, t_{\ell+1})$ with high probability before the bad round $s(a)$ occurs. Going forward, to simplify notation we will drop the dependence on the fixed arm $a$ in some variables.

For each bad segment $[s_{i,j}(a), s_{i,j+1}(a))$, recall that $\tilde{s}_{i,j}(a)$ is the approximate midpoint between $s_{i,j}(a)$ and $s_{i,j+1}(a)$ (see Definition 6). Next, let $m_{i,j} := 2^n$ where $n \in \mathbb{N}$ satisfies:

$$2^n \geq s_{i,j+1}(a) - s_{i,j}(a) > 2^{n-1}.$$

Plainly, $m_{i,j}$ is a dyadic approximation of the bad segment length. Next, recall that the Bernoulli $B_{t,m}$ decides whether $\mathsf{Base\text{-}Alg}(t, m)$ is scheduled at round $t$ (see Line 6 of Algorithm 1). If for

some $t \in [s_{i,j}(a), \tilde{s}_{i,j}(a)]$, $B_{t,m_{i,j}} = 1$, i.e. a perfect replay is scheduled, then $a$ will be evicted from $\mathcal{A}_{\text{master}}$ by round $s_{i,j+1}(a)$ (Proposition 14). We will show this happens with high probability via concentration on the sum

$$S(a, t_\ell) := \sum_{(i,j):s_{i,j+1}(a)<s(a)} \sum_{t=s_{i,j}(a)}^{\tilde{s}_{i,j}(a)} B_{t,m_{i,j}},$$

Note that the random variable $S(a, t_\ell)$ only depends on the replay scheduling probabilities $\{B_{s,m}\}_{s,m}$ given a fixed arm $a$ and episode start time $t_\ell$, since the bad round $s(a)$ is also fixed given these quantities. This means that $S(a, t_\ell)$ is an independent sum of Bernoulli random variables $B_{t,m_{i,j}}$, conditional on $t_\ell$. Then, a Chernoff bound over the randomness of $S(a, t_\ell)$, conditional on $t_\ell$ yields

$$\mathbb{P}\left(S(a, t_\ell) \leq \frac{\mathbb{E}[S(a, t_\ell) \mid t_\ell]}{2} \mid t_\ell\right) \leq \exp\left(-\frac{\mathbb{E}[S(a, t_\ell) \mid t_\ell]}{8}\right).$$

The above RHS error probability is bounded above above by $1/T^3$ by observing:

$$\mathbb{E}\left[S(a, t_\ell) \mid t_\ell\right] \geq \sum_{(i,j)} \sum_{t=s_{i,j}(a)}^{\tilde{s}_{i,j}(a)} \frac{1}{\sqrt{m_{i,j} \cdot (t - t_\ell)}} \geq \frac{1}{4} \sum_{(i,j)} \sqrt{\frac{s_{i,j+1}(a) - s_{i,j}(a)}{s(a) - t_\ell}} \geq \frac{c_6}{4} \log(T),$$

for $c_6 > 0$ large enough, where the last inequality follows from (17) in the definition of the bad round $s(a)$ (Definition 7). Taking a further union bound over the choice of arm $a \in [K]$ gives us that $S(a, t_\ell) > 1$ for all choices of arm $a$ (define this as the good event $\mathcal{E}_2(t_\ell)$) with probability at least $1 - K/T^3$. This means arm $a$ will be evicted before round $s(a)$ with high probability.

Recall on the event $\mathcal{E}_1$ the concentration bounds of Proposition 10 hold. Then, on $\mathcal{E}_1 \cap \mathcal{E}_2(t_\ell)$, letting $a = a_\ell$ in the preceding arguments we must have $t_{\ell+1} - 1 \leq s(a_\ell)$ Thus, by the definition of the bad round $s(a_\ell)$ (Definition 7), we must have:

$$\sum_{[s_{i,j}(a_\ell), s_{i,j+1}(a_\ell)):s_{i,j+1}(a_\ell)<t_{\ell+1}-1} \sqrt{s_{i,j+1}(a_\ell) - s_{i,j}(a_\ell)} \leq c_6 \log(T)\sqrt{t_{\ell+1} - t_\ell}. \tag{18}$$

Thus, by (14) in the definition of bad segments (Definition 5), over the bad segments $[s_{i,j}(a_\ell), s_{i,j+1}(a_\ell))$ which elapse before the end of the episode $t_{\ell+1} - 1$, the regret of $a_\ell$ to $a_t^\sharp$ is at most order $\log^2(T)\sqrt{K \cdot (t_{\ell+1} - t_\ell)}$.

Over each non-bad critical segment $[s_{i,j}(a_\ell), s_{i,j+1}(a_\ell))$, the regret of playing arm $a_\ell$ to $a_i^\sharp$ is at most $\log(T)\sqrt{\tau_{i+1} - \tau_i}$ since there is at most one non-bad critical segment per phase $[\tau_i, \tau_{i+1})$ (follows from Definition 5).

So, we conclude that on event $\mathcal{E}_1 \cap \mathcal{E}_2(t_\ell)$:

$$\sum_{t=t_\ell}^{t_{\ell+1}-1} \delta_t(a_t^\sharp, a_\ell) \leq c_7 \log^2(T) \sum_{i \in \text{PHASES}(t_\ell, t_{\ell+1})} \sqrt{K(\tau_{i+1} - \tau_i)}.$$

Taking expectation, we have by conditioning first on $t_\ell$ and then on event $\mathcal{E}_1 \cap \mathcal{E}_2(t_\ell)$:

$$\mathbb{E}\left[\sum_{t=t_\ell}^{t_{\ell+1}-1} \delta_t(a_t^\sharp, a_\ell)\right] \leq \mathbb{E}_{t_\ell}\left[\mathbb{E}\left[\mathbf{1}\{\mathcal{E}_1 \cap \mathcal{E}_2(t_\ell)\} \sum_{t=t_\ell}^{t_{\ell+1}-1} \delta_t(a_t^\sharp, a_\ell) \mid t_\ell\right]\right]$$
$$+ T \cdot \mathbb{E}_{t_\ell}\left[\mathbb{E}\left[\mathbf{1}\{\mathcal{E}_1^c \cup \mathcal{E}_2^c(t_\ell)\} \mid t_\ell\right]\right]$$
$$\leq c_7 \log^2(T)\mathbb{E}_{t_\ell}\left[\mathbb{E}\left[\mathbf{1}\{\mathcal{E}_1 \cap \mathcal{E}_2(t_\ell)\} \sum_{i \in \text{PHASES}(t_\ell, t_{\ell+1})} \sqrt{K(\tau_{i+1} - \tau_i)} \mid t_\ell\right]\right]$$
$$+ \frac{2K}{T^2}$$
$$\leq c_7 \log^2(T)\mathbb{E}\left[\mathbf{1}\{\mathcal{E}_1\} \sum_{i \in \text{PHASES}(t_\ell, t_{\ell+1})} \sqrt{\tau_{i+1} - \tau_i}\right] + \frac{2}{T},$$

where in the last step we bound $\mathbf{1}\{\mathcal{E}_1 \cap \mathcal{E}_2(t_\ell)\} \leq \mathbf{1}\{\mathcal{E}_1\}$ and apply tower law again. This concludes the proof. ∎

