# OpenReview forum: "When Can We Track Significant Preference Shifts in Dueling Bandits?"
_NeurIPS.cc/2023/Conference — NeurIPS 2023 poster_

### Official Review · Reviewer_VW3L · 2023-07-03

**Soundness:** 4 excellent
**Presentation:** 3 good
**Contribution:** 3 good
**Rating:** 6
**Confidence:** 3

**Summary:**

This paper studies the non-stationary multi-armed dueling bandit under the SST and STI conditions. The paper first shows a hardness result that SST+STI are necessary for sub-linear regret, while merely SST or STI cannot guarantee sublinear regrets. Then, the paper studies under SST+STI conditions and propose a new adaptive algorithm that achieves $\tilde{O}(\sqrt{\tilde{L}KT})$ regret. This regret upper bound improves upon the previous one $\tilde{O}(K\sqrt{\tilde{L}T})$.

**Strengths:**

The paper is well-written and relatively easy for me to follow. The lower bound result is interesting in that it reveals that the Condorcet winner condition is not sufficient for efficient learning in the non-stationary environment. Instead, both SST and STI must hold to allow a sub-linear regret. For the upper bound, a new algorithm along with its analysis is provided to achieve an improved regret upper bound. I find the explanation and reasoning for such an algorithm framework to be quite informative. The analysis of why previous works don't have better dependence on $K$ (Section 5.1) shows the algorithm design is original.

**Weaknesses:**

1. While the impossibility result shows that SST+STI are necessary, it is not clear what is the minimax lower bound under such conditions. Is it $\Omega(\sqrt{LKT})$ or $\Omega(\sqrt{ \tilde{L}KT})$? It is not discussed in this work what is the lower bound in terms of significant shifts. Therefore, it is not well-supported to claim the upper bound is optimal.


**Questions:**

1. As mentioned in the section above. What is the lower bound for the setting in this paper? In the paper, it is mentioned that one lower bound is $\Omega(\sqrt{LKT})$, while your upper bound is $\Omega(\sqrt{\tilde{L}KT})$ with $\tilde{L} << L$. This is contradicting without some context.

2. There are a series of related works discussing ranking with/without SST or STI conditions. Could you also comment on these works:

    * On Sample Complexity Upper and Lower Bounds for Exact Ranking from Noisy Comparisons (NeurIPS'19) Ren et al.
    * Active Ranking without Strong Stochastic Transitivity (NeurIPS'22) Lou et al.



**Limitations:**

N/A due to its theoretical nature.

---

> ### Author Rebuttal · Authors · 2023-08-10
>
> * _Lower Bound and $L$ vs. $\tilde{L}$_: the lower bound is in fact $\sqrt{\tilde{L} KT}$ since the previous work’s lower bound construction in terms of $L$ switches in fact uses $\tilde{L}=L$ (Theorem 5.1; Saha and Gupta, 2022). This is also stated in Remark 3.1 of the cited paper Saha and Buening (2022). We apologize for any confusion about this and will make this more explicit. Thus, our upper bounds are optimal up to log factors.
> * _Works on Ranking with/without SST/STI_: thanks for the mentioned works on ranking, which we’ll cite and discuss in the final version. We do note that these works are not in online bandit settings. Hence, the techniques are not directly applicable to our problem of dueling bandits with distribution shifts.

---

> > ### Comment · Reviewer_VW3L · 2023-08-10
> > **Thanks**
> >
> > Thank you for the response. My concerns are addressed.

---

### Official Review · Reviewer_6oeV · 2023-07-05

**Soundness:** 3 good
**Presentation:** 3 good
**Contribution:** 3 good
**Rating:** 6
**Confidence:** 3

**Summary:**

The paper tackles a regret minimizing problem in a dueling bandits scenario, where the underlying preference probabilities are non-stationary but instead shifting over time. In case these preference probabilities fulfill SST (strong stochastic transitivity) and STI (stochastic triangle inequality) at any time, the authors present a novel upper bound on the dynamic regret that depends on the (unknown) number of so-called significant shifts, the number of arms and the time horizon. Moreover, they provide evidence that leaving out one of the assumptions SST and STI would result in a learning scenario where suffering linear regret is in a worst case unavoidable.

**Strengths:**

This paper presents a novel algorithm for the non-stationary dueling bandits problem, which is able to learn without knowing the number of significant shifts. It comes with an upper bound on the dynamic regret. This bound and the hardness result on the learnability in case one of SST and STI are violated (Thm 3) are highly non-trivial results. Mostly, the paper and proofs are written in a convenient way.

**Weaknesses:**

- Unfortunately, this paper does not provide any experiments. Why not?
- You say (e.g. in your conclusion and in 174ff.) that one cannot achieve a dynamic regret of order $O(\sqrt{K\tilde{L}T})$ outside of $SST \cap STI$. However, you only prove that non-linear dynamic regret can not be avoided in a worst-case sense (!) under $STI \setminus SST$ and under $SST \setminus STI$. This (a) does not formally imply any worst-case hardness result in case none of SST and STI is fulfilled, right?, and (b) does not exclude existence of a class X' outside $SST\cap STI$ where  $O(\sqrt{K\tilde{L}T})$ is achievable.
- 162: The constant C in (3) is required to implement and execute Algorithm 2. Do you already provide it somewhere? If so, please state it here, otherwise, provide it.
- 516: $c_1$ is not given, so $\mathcal{E}_1$ is not well-defined.
- Sec. 3 is called 'Hardness of significant shifts...'. However, Thm. 3 shows a negative result for a scenario with no significant shifts. How are the section title and Thm. 3 related?

Minor remarks, questions and typos:
- 135: Why do you write $\succeq$ instead of $\succ$?
- 196: It seems as if METASWIFT has not been properly introduced yet.
- 470: KL [divergence]
- 477: ")" missing
- 488: observations
- 506: Sec. 6?
- 506: a_t^\# is defined in Sec. 6, but not in the App. B
- 536: Sec. 6
- In Algo 2: Refer for the definition of $B_{s,m}$ to Algo 1

**Questions:**

- You have proven in Thm. 4 an upper regret bound. What could you say about a lower bound? How sharp is your bound?
- 162: Can Thm. 3 be generalized to hold for fixed K>3?

**Limitations:**

The authors adequately addressed the limitations of their work.

---

> ### Author Rebuttal · Authors · 2023-08-10
>
> Thanks for the close reading, pointing out typos, and useful writing suggestions.
> * _On experiments_: please see the general author rebuttal.
> * _$\sqrt{K}$ Regret in Other Subclasses_: indeed there may be carefully constructed subclasses/instances outside of $SST\cap STI$ where $\sqrt{\tilde{L} KT}$ rates are possible. We will make this point clear in the final version of our paper. In our work we focus on commonly studied classes such as STI, SST, Condorcet Winner, Copeland Winner etc. Our result disallows such classes including STI (by itself), SST (by itself), Condorcet Winner and Copeland Winner. It is standard in the dueling bandit literature to consider classes of problems which are either subsumed by SST and STI or contain them.
> * _Constant $C$_: $C$ is a fixed constant whose value can be determined from the regret analysis of Theorem 4 (i.e., we may take $C=1+10\cdot (e-1)$ formally). We apologize for any confusion about this and will make this more explicit.
> * _$c_1$ and_ $\mathcal{E}_1$: $c_1$ is in fact a fixed constant whose value is determined by Lemma 9 and the proof of Prop 10 (i.e., $c_1=10\cdot (e-1)$ formally). In many places in the proof, we do not make constants such as $c_1,c_2,\ldots$ explicit for ease of presentation. We apologize for any confusion this may have caused.
> * _Lower Bound and Tightness of Upper Bound_: There is a lower bound of order $\sqrt{\tilde{L} KT}$ in the previous work (cf. Saha and Gupta, 2022; Theorem 5.1). This means our upper bound is in fact tight up to log factors.
> * _Generalizing Thm 4 for $K>3$_: Yes, Thm 3 can be generalized for K>3. For instance, there can be any fixed number of arms which randomly switch between being the winner and the unsafe arm in the construction of ${\bf P}^+, {\bf P}^-$ in the proof of Thm 3.

---

> > ### Comment · Reviewer_6oeV · 2023-08-15
> >
> > Thank you for your response, you have addressed all my concerns. I raised my score.

---

### Official Review · Reviewer_W1nd · 2023-07-07

**Soundness:** 4 excellent
**Presentation:** 3 good
**Contribution:** 4 excellent
**Rating:** 7
**Confidence:** 3

**Summary:**

This work studies the problem of a non-stationary bandits problem. First of all, the hardness of dynamic regret minimization under significant shifts problem without SST+STI condition is discussed. Then, with such additional conditions, an algorithm is proposed with a regret of $O(\sqrt{KLT})$. An innovative arm selection strategy which maintains a set of low regret arms is used in replacement of those greedily picking the best arm so far in static environments.

**Strengths:**

The proposed method differentiates itself with conventional arm selection strategy in stationary environments  by maintaining an acceptable level of regret of order $O(\sqrt{KT})$ and simultaneously monitoring the change of the environment.  This work may have significant use case compared to stationary algorithms given the fact the in real-life scenarios the environment is always changing.

**Weaknesses:**

Simulation study on two cases might deepen the understanding of the proposed method: Although orderwise the same, a comparison between stationary method and this work under static settings can reveal the overhead, and a comparison of the existing work such as [10] with the proposed method should verify the K factor. Given this line of research usually does not carry empirical result and is of theoretical nature. This is just a nice to have suggestion.

**Questions:**

See weakness.

**Limitations:**

The limitations are well discussed in the paper.

---

> ### Author Rebuttal · Authors · 2023-08-10
>
> Thanks for your positive comments. Please see simulation results in the general author rebuttal.

---

> > ### Comment · Reviewer_W1nd · 2023-08-12
> >
> > Thanks for the response. You have addressed my concerns.

---

### Official Review · Reviewer_FDu2 · 2023-07-20

**Soundness:** 3 good
**Presentation:** 3 good
**Contribution:** 3 good
**Rating:** 6
**Confidence:** 3

**Summary:**

This paper addresses the problem of dueling bandits with distribution shifts in preferences. The authors investigate the possibility of designing an adaptive algorithm with dynamic regret that depends on the number of significant shifts in preferences. They provide an impossibility result for well-studied preference distributions and show that it is possible to design such an algorithm under certain conditions. The paper introduces novel algorithmic approaches, including the SWIFT and METASWIFT algorithms, and contributes to the understanding of regret bounds in switching dueling bandits. The authors also discuss the difficulty of efficient exploration in non-stationary dueling bandit problems and suggest the need for smarter exploration strategies to achieve optimal regret bounds. Overall, the paper presents theoretical results, algorithmic solutions, and future research directions in the field of dueling bandits.

**Strengths:**

This paper introduces the novel problem of dueling bandits with distribution shifts in preferences, and proposes a unique adaptive algorithm with dynamic regret that accounts for the number of significant shifts in preferences. The authors provide a thorough theoretical analysis, rigorous proofs, and well-designed algorithmic approaches, such as the SWIFT and METASWIFT algorithms. The paper is well-written, organized, and presents clear explanations and detailed pseudocode, making it easy for readers to understand and replicate the experiments. The authors’ contributions advance the state-of-the-art in dueling bandits and provide valuable insights for researchers and practitioners. Overall, this paper demonstrates originality, quality, clarity, and significance in its contributions to the field of computer science.

**Weaknesses:**

The paper would benefit from the inclusion of empirical evaluation to complement its theoretical analysis.  The authors should conduct extensive experiments on a wider range of datasets and problem instances to validate the performance of the proposed algorithms in practical scenarios.  Additionally, it would be valuable for the authors to provide insights into the computational complexity and scalability of the algorithms, as these are important aspects for assessing the practical relevance of their work.

Another weakness is the unclear explanation of the innovation in theoretical analysis. The authors mentioned in section 1.1 the innovation of their algorithm compared to previous work, but did not discuss how their theoretical analysis differs from existing techniques, especially in the proof analysis where some techniques from [10] and [30] were used. The authors should clearly explain the innovation in theoretical analysis and highlight the novelty and importance of their contributions in this aspect.


**Questions:**

Please refer to weaknesses.

**Limitations:**

The paper does not explicitly discuss the limitations of the proposed algorithms or the potential negative consequences that may arise from their application.

---

> ### Author Rebuttal · Authors · 2023-08-10
>
> Thanks for encouraging comments!
> * _On experiments_: please see preliminary experimental results in the general author rebuttal.
> * _Innovation in analysis_: the key innovation is explained in Section 5.1 of the paper. At first glance, the analysis appears challenging as optimal dueling algorithms (e.g., Beat-the-Mean, Interleaved Filtering, Borda scores) use elimination rules and regret analyses which crucially depend on there being a fixed ranking of arms, so that, at any time, lowest-ranked arms can be reliably and quickly eliminated. However, in our **non-stationary** setting, this order-dependent analysis is no longer possible due to a significant new challenge: the ordering over arms can change every single round without triggering a significant shift. How can one design an elimination rule in an order-independent manner? This appears challenging because the goal of minimizing regret and finding an approximately correct ordering seem to be tied together. But, actually our regret analysis shows that finding an ordering is not necessary for the goal of regret minimization. We give a new elimination rule and regret analysis that is independent of any ordering over arms. It is surprising that such an elimination rule/analysis exists given the prevalence of order-based algorithms discussed above. We believe that our elimination rule will be useful in the dueling bandits literature beyond this current work. We apologize for not making this innovation clear elsewhere in the paper and will fix the writing. We would also like to point out that our lower bound is novel and non-trivial, and discovers interesting insights about the problem.

---

> > ### Comment · Reviewer_FDu2 · 2023-08-11
> >
> > Thank you for your response. I have thoroughly reviewed your response as well as the comments provided by other reviewers. After careful consideration, I have decided to maintain my current score.

---

### Author Rebuttal · Authors · 2023-08-10

We thank reviewers for their time and careful reading.

A common concern raised by reviewers was the lack of empirical experiments. We'd like to emphasize that the main contribution of the paper is a theoretical understanding of achievability: whether $\sqrt{K}$ dependence in regret is possible in dueling bandits with significant shifts and under what models of pairwise comparisons. Our contribution is consistent with the broader literature on "bandits with distribution shifts" which has been largely theoretical in nature. We agree with reviewers, however, that an eventual goal of this emerging line of work is to develop practical procedures.

As a step in that direction, we provide preliminary experimental results here, which will be expanded in the final version of the paper if accepted.

**Preliminary Experimental Results:** (see attached pdf)
The first figure on the left shows the regret curves in a stationary environment with $S=0$ changes. The second figure on the right shows the regret curves in a non-stationary environment with $S=4$ changes. Exact mean and standard deviations on final regret are given in the table. These do **support the theoretical message** that METASWIFT performs better than the existing ANACONDA algorithm in non-stationary environments due to more efficient exploration of arms (demonstrated through $\sqrt{K}$ versus $K$ dependence in the theoretical bounds). Moreover, we also observe that the IF algorithm which is designed for stationary environments can have almost linear regret in non-stationary environments.

Further details on experiments are as follows:

**Environments**: were generated using a geometric BTL model where the arms are linearly ordered and the $i$-th best arm beats the $j$-th best arm with probability $\frac{2^{-i}}{2^{-i}+2^{-j}}$. At each changepoint, the ordering of arms was randomly permuted. The horizon was set to $T=50000$ with $K=10$ arms. Regret was computed over $N=50$ trials of each environment and standard confidence bands are depicted in the figures.

**Algorithms**: we considered four algorithms: (1) our algorithm METASWIFT, (2) the ANACONDA algorithm of Buening & Saha 2022, (3) Interleaved Filtering (which we abbreviate as IF) as specified by Yue et al, 2009 for the SST$\cap$STI model, and a baseline (3) RANDDUEL which naively plays a pair of arms selected uniformly randomly at every round.

**Parameters**: parameters associated with each of the algorithms (e.g., the constants in displays (3) and (4) of our submission, analogous quantities in ANACONDA, and IF's eliminination threshold) were tuned using cross validation on randomly generated geometric BTL environments as described above with number of changepoints varying from $0$ to $1000$. For fairness, all algorithms were given the chance to tune parameters on the same environments before testing.

Please see indvidual rebuttals to each reviewer for further responses.

---

### Decision · Program_Chairs · 2023-09-21

**Decision:**

Accept (poster)

**Comment:**

The paper studies dueling bandits with distribution shifts, and provides an adaptive algorithm with a regret analysis. All reviewers show positive support to the paper, with scores, 7, 6, 6, 6. The reviewers acknowledged the novelty of the study, and the authors rebuttal further helped clarifying some of the issues raised by the reviewer. Due to the unanimous positive support on the paper, I recommend acceptance to the paper, and encourage the authors to provide a thorough revision to incorporate all comments from the reviewers.